

**A multi-year observation of nitrous oxide at the Boknis Eck**
**Time-Series Station in the Eckernförde Bay (southwestern**
**Baltic Sea)**
Xiao Ma[1], Sinikka T. Lennartz[1,2], and Hermann W. Bange[1]
[1] GEOMAR Helmholtz Centre for Ocean Research Kiel, Düsternbrooker Weg 20, 24105 Kiel, Germany
[2] now at ICBM, University of Oldenburg, Oldenburg, Germany
*Correspondence to:* Xiao Ma (mxiao@geomar.de)
**Abstract.** Nitrous oxide ($N_2O$) is a potent greenhouse gas and it is involved in stratospheric ozone
depletion. Its oceanic production is mainly influenced by dissolved nutrient and oxygen ($O_2$)
concentrations in the water column. Here we examined the seasonal and annual variations of
dissolved $N_2O$ at the Boknis Eck (BE) Time-Series Station located in Eckernförde Bay
(southwestern Baltic Sea). Monthly measurements of $N_2O$ started in July 2005. We found a
pronounced seasonal pattern for $N_2O$ with high concentrations (supersaturations) in winter/early
spring and low concentrations (undersaturations) in autumn when hypoxic/anoxic conditions
prevail. Unusually low $N_2O$ concentrations were observed during October 2016–April 2017,
which was presumably a result of prolonged anoxia and the subsequent nutrient deficiency.
Unusually high $N_2O$ concentrations were found in November 2017 and this event was linked to
the occurrence of upwelling which interrupted $N_2O$ consumption via denitrification and
potentially promoted ammonium oxidation (nitrification) at the oxic/anoxic interface. Nutrient
concentrations (such as nitrate, nitrite and phosphate) at BE are decreasing since 1980s, but
oxygen concentrations in the water column are still decreasing. Our results indicate a close
coupling of $N_2O$ anomalies to $O_2$ concentration, nutrients and stratification. Given the long-term
trends of declining nutrient and oxygen concentrations at BE, a decrease in $N_2O$ concentration,
and thus emissions, seems likely due to an increasing number of events with low $N_2O$
concentrations.

## 1. Introduction

Long-term observation with regular measurement intervals can be an effective way to monitor
seasonal and interannual variabilities as well as to decipher short- and long-term trends of an
ecosystem, which are required to make projections of the future ecosystem development (see e.g.
Ducklow et al., 2009). Recently, multi-year time-series measurements of nitrous oxide ($N_2O$), a
potent greenhouse gas and a major threat to ozone depletion (IPCC, 2013; Ravishankara et al.,
2009), have been reported from the coastal upwelling areas off central Chile (Farías et al., 2015)
and off Goa (Naqvi et al., 2010), in the North Pacific Subtropical Gyre (Wilson et al., 2017), and
in Saanich Inlet (Capelle et al., 2018).



N$_2$O production in the ocean is generally dominated by microbial nitrification (NH$_4^+$ → NO$_2^-$ →
NO$_3^-$) and denitrification (NO$_3^-$ → NO$_2^-$ → N$_2$O → N$_2$). During bacterial/archaeal nitrification,
N$_2$O is produced as a by-product with enhanced N$_2$O production under low oxygen (O$_2$)
conditions (e.g. Goreau et al., 1980; Löscher et al., 2012). N$_2$O is produced as an intermediate
during bacterial denitrification (Codispoti et al., 2005). N$_2$O could be further consumed via
denitrification to dinitrogen, however, this process is inhibited with the presence of O$_2$ because
of the low O$_2$ tolerance of the enzyme involved (Bonin et al. 1989). This incomplete pathway is
called partial denitrification and can lead to N$_2$O accumulation (e.g. Naqvi et al., 2000; Farías et
al., 2009).
The oceans including coastal areas contribute ~25% of the natural and anthropogenic N$_2$O
emissions (IPCC, 2013), with disproportionately high emissions from coastal and estuarine areas
(Bange, 2006). N$_2$O emissions from coastal areas strongly depend on nitrogen inputs (Seitzinger
and Kroeze, 1998; Zhang et al., 2010). The increasing input of nitrogen (i.e. eutrophication) has
become a worldwide problem in coastal waters leading to enhanced productivity and severe O$_2$
depletion caused by enhanced degradation of organic matter (Breitburg et al., 2017; Rabalais et
al., 2014). The decline in O$_2$ concentration (i.e. deoxygenation), either in coastal waters or the
open ocean, might result in favorable conditions for N$_2$O production (Codispoti et al., 2001;
Nevison et al., 2003). The results of a model study by Kroeze and Seitzinger (1998) indicated a
significant increase of N$_2$O in European coastal waters for 2050. Moreover, it has been suggested
that N$_2$O production and emissions are very likely to increase in the near future, especially in the
shallow suboxic/anoxic coastal systems (Naqvi et al., 2000; Bange, 2006). However, model
projections show a net decrease in future global oceanic N$_2$O emission during the 21$^{st}$ century
(Martinez-Rey et al., 2015; Landolfi et al., 2017; Battaglia and Joos, 2018).
The Baltic Sea is a nearly enclosed, marginal sea with a very limited access to the open ocean via
the North Sea. The restricted water exchange with the North Sea and extensive human activities,
such agriculture, industrial production and sewage discharge in the catchment area led to high
inputs of nutrients to the Baltic Sea. As a result, the areas affected by anoxia have been
expanding in the deep basins of the central Baltic Sea (Carstensen et al., 2014). In order to
control this situation, the Helsinki Commission (HELCOM) was established in 1974 and a series
of measures have been taken to prevent anthropogenic nutrient input into the Baltic Sea.
Consequently, the nutrient inputs (by riverine loads, direct point-sources and, for nitrogen,
atmospheric deposition) to the Baltic Sea are declining (HELCOM, 2018a). However, the
number of low O$_2$ (i.e. hypoxic/anoxic) events in coastal waters of the Baltic Sea is increasing
and deoxygenation is still going on (Conley et al., 2011; Lennartz et al., 2014). The
deoxygenation in the Baltic Sea can affect the production/consumption of N$_2$O. Our group has
been monitoring dissolved N$_2$O concentrations at the Boknis Eck Time-Series Station, located in
Eckernförde Bay (southwestern Baltic Sea), for more than a decade. In this study, we present
monthly measurements of N$_2$O and biogeochemical parameters such as nutrients and O$_2$ from
July 2005 to December 2017. The major objectives of our study were: 1) to decipher the seasonal



pattern of N$_2$O distribution in the water column, 2) to identify short-term and long-term trends of
the N$_2$O concentrations, 3) to explore the potential role of nutrients and O$_2$ for N$_2$O
production/consumption, and 4) to quantify the sea–to–air N$_2$O flux density at the time-series
station.

## 2. Material and methods

### 2.1 Study site

Sampling at the Boknis Eck (BE) Time-Series Station (www.bokniseck.de) started on 30 April
1957 and, therefore, it is one of the oldest continuously operated time-series stations in the world.
The BE station is located at the entrance of the Eckernförde Bay (54°31′ N, 10°02′ E, Fig. 1) in
the southwestern Baltic Sea. The water depth of the sampling site is 28 m. Various physical,
chemical and biological parameters are measured on a monthly basis (Lennartz et al., 2014).
There is no significant river runoff to Eckernförde Bay and the saline North Sea water inflow
from the Kattegat plays a dominant role for the local hydrographic conditions. Seasonal
stratification usually starts to develop in April and lasts until October, during which hypoxia or
even anoxia (characterized by the presence of hydrogen sulphide, H$_2$S) sporadically occurs, as a
result of restricted vertical water exchange and bacterial decomposition of organic matter in the
bottom water (Hansen et al., 1999; Lennartz et al., 2014). Thus, BE is a natural laboratory to
study the influence of O$_2$ variations and anthropogenic nutrient loads on N$_2$O
production/consumption.

### 2.2 Sample collection and measurement

Monthly sampling of N$_2$O at the BE Time-Series Station started in July 2005. Triplicate samples
were collected from six depths (1, 5, 10, 15, 20 and 25 m). Seawater was drawn from 5 L Niskin
bottles into 20 mL brown glass vials after overflow. The vials were sealed with rubber stoppers
and aluminum caps. The bubble-free samples were poisoned with 50 µL of a saturated mercury
chloride (HgCl$_2$) solution and then stored in a cool, dark place until measurement. The general
storage time before measurements of the N$_2$O concentrations was less than three months.
The static headspace-equilibrium method was adopted to measure the dissolved N$_2$O
concentrations in the vials. 10 mL helium (99.9999 %, AirLiquide, Düsseldorf, Germany)
headspace was created in each vial with a gas-tight glass syringe (VICI Precision Sampling,
Baton Rouge, LA). Samples were vibrated with Vortex (G-560E, Scientific Industries Inc., New
York, USA) for 20 seconds and then left for at least two hours until equilibrium. 9.5 mL
subsample of the headspace was subsequently injected into a GC-ECD (gas chromatograph
equipped with the electron capture detector) system (Hewlett-Packard 5890 Series II, Agilent
Technologies, Santa Clara, CA, USA), which was calibrated with two standard gas mixtures
(N$_2$O in synthetic air, Deuste-Steininger GmbH, Mühlhausen, Germany and Westfalen AG,
Münster, Germany) prior to the measurement. The average precision of the measurements,



calculated as the median standard deviation from triplicate measurements, was 0.4 nM.
Triplicates with a standard deviation of >10% were omitted. More details about the $N_2O$
measurement can be found in Kock et al. (2016). Dissolved oxygen ($O_2$) concentrations were
measured by Winkler titrations (Grasshoff et al., 1999). Nutrient concentrations were measured
by the Segmented Continuous Flow Analysis (SCFA, Grasshoff et al., 1999). A more detailed
summary of the parameters measured and methods applied can be found in Lennartz et al. (2014).

### 2.3 Times series analysis

A time-series can be decomposed into three main components, i.e. trend, cycle and residual
component (Schlittgen and Streitberg, 2001). We used the Mann–Kendall test and wavelet
analysis to detect the trend and periodical cycles in the time-series data, respectively. As for the
residual component, we highlight unusual high/low $N_2O$ concentrations during 2005-2017 and
discuss the potential causes for these events.

### 2.3.1 Wavelet analysis

In order to decipher periodical cycles of the parameters collected at the BE Time-Series Station,
a wavelet analysis method was adopted. Wavelet analysis enables the detection of the period and
the temporal occurrence of repeated cycles in time-series data. One of the requirements for
wavelet analysis is a regular, continuous time-series. Since there is data missing (maximum 2
months in a row) in the BE time-series, due to terrible weather or the ship's unavailability,
missing data was interpolated from the previous and following months. Data was shifted to the
15$^{th}$ of each month to obtain regular spacing. Considering the band width in both frequency and
time domain, a Morlet mother wavelet with a wave number of 6 was chosen (Torrence and
Compo, 1998). The mother wavelet was then scaled between the frequency of a half-year cycle
and the length of the time-series with a stepsize of 0.25. The wavelet analysis was conducted
with the MatLab code by Torrence and Compo [2004]. More information about the method can
be found on the website http://paos.colorado.edu/research/wavelets/.

### 2.3.2 Mann–Kendall test

Mann–Kendall test (MKT) is a non-parametric statistical test to assess the significance of
monotonic trends for time-series measurements. It tests the null hypothesis that all variables are
randomly distributed against the alternative hypothesis that a monotonic trend, either increase or
decrease, exists in the time-series on a given significance level α (here α=0.05). MKT is flexible
for data with missing values and the results are not impacted by the magnitude of extreme values,
which makes it a widely used test in hydrology and climatology (e.g. Xu et al., 2003; Yang et al.,
2004). However, MKT is sensitive to serial correlation in the time-series. The presence of
positive serial correlation would increase the probability of trend detection even though no such
trend exists (Kulkarni and von Storch, 1995). In order to avoid this situation, data from 12
months were tested individually. It is assumed that there is no residual effect left from the same



month last year, considering that the nitrogen species are rapidly biologically cycled. The Matlab
function from Simone (2009) was used for the MKT.
**2.4 Calculation of saturation and sea–to–air flux density**
$N_2O$ saturations ($S_{N2O}$, %) were calculated as:
$$S_{N2O} = 100 \times N_2O_{obs}/N_2O_{eq} \qquad (1)$$

where $N_2O_{obs}$ and $N_2O_{eq}$ (in nM) are the observed and equilibrated $N_2O$ concentrations in
seawater, respectively. $N_2O_{eq}$ was computed as a function of surface seawater temperature, in
situ salinity (Weiss and Price, 1980) and the dry mole fractions of atmospheric $N_2O$ at the time
of the sampling. The dry mole fractions of atmospheric $N_2O$ were derived from the monthly
average of $N_2O$ data measured at Mace Head, Ireland (AGAGE, http://agage.mit.edu/).
The excess of $N_2O$ ($\Delta N_2O$) and apparent oxygen utilization (AOU) was calculated as:
$$\Delta N_2O = N_2O_{obs} - N_2O_{eq} \qquad (2)$$

$$AOU = O_{2eq} - O_{2obs} \qquad (3)$$

where $O_{2eq}$ and $O_{2obs}$ (in μM) are the equilibrated and observed $O_2$ concentrations in seawater,
respectively. The equilibrated $O_2$ concentrations were calculated with the equation given by
Weiss (1970).
$N_2O$ flux density ($F_{N2O}$, in nmol m$^{-2}$ s$^{-1}$) was calculated as:
$$F_{N2O} = k_{N2O} \times (N_2O_{obs} - N_2O_{eq}) \qquad (4)$$

where $k_{N2O}$ (in cm h$^{-1}$) is the gas transfer velocity calculated with the method given by
Nightingale et al. (2000), as a function of the wind speed and the Schmidt number (Sc). The
wind speed data were obtained from the Kiel Lighthouse (see: www.geomar.de/service/wetter/),
which is approximately 20 km away from the BE Time-Series Station. The wind speed was
normalized to 10 m ($u_{10}$) to calculate $k_{N2O}$ (Hsu et al., 1994). $k_{N2O}$ was adjusted by multiplying
with (Sc/600)$^{-0.5}$, and Sc was computed according to Walter et al. (2004).
**3. Result and discussion**
**3.1 Overview**
$N_2O$ concentrations at the BE Time-Series Station showed significant temporal and depth-
dependent variations from 2005 to 2017 (Fig. 2). $N_2O$ concentrations fluctuated between 1.2 and
37.8 nM, with an overall average of 13.9±4.2 nM. This value was higher than the results from
the surface water of Station ALOHA (5.9–7.4 nmol kg$^{-1}$, average 6.5±0.3 nmol kg$^{-1}$, Wilson et
al., 2017), which is reasonable considering the weak anthropogenic impact in the North Pacific



Subtropical Gyre. The $N_2O$ concentrations at BE were much lower than those measured at the time-series station in the coastal upwelling area off Chile (2.9–492 nM, average 39.4±29.2 nM in the oxyclines and 37.6±23.3 nM in the bottom waters, Farías et al., 2015) and a quasi-time series station off Goa (Naqvi et al., 2010), where significant $N_2O$ accumulations were observed in subsurface waters at both locations. Our measurements were comparable to the time-series station from Saanich Inlet (~0.5–37.4 nM, average 14.7 nM, Capelle et al., 2018), a seasonally anoxic fjord which has similar hydrographic conditions as BE.

$NO_2^-$ concentrations fluctuated between below detection limit of 0.1 µM and 1.6 µM, with an average of 0.2±0.3 µM. $NO_3^-$ concentrations varied from below detection limit of 0.3 µM to 17.9 µM, with an average of 2.0±2.8 µM. The temporal and spatial distributions of nitrite ($NO_2^-$) and nitrate ($NO_3^-$) were similar during 2005–2017. A clear $O_2$ seasonality can be seen with severe $O_2$ depletion in the bottom waters during summer and autumn. Anoxia with the presence of $H_2S$ were detected in September/October 2005, September 2007, September/October 2014, and September–November 2016. All of the extremely low $N_2O$ concentrations (<5 nM) were observed in the bottom waters in autumn, coinciding with hypoxia/anoxia, while the high $N_2O$ concentrations (>20 nM) sporadically occurred at different depths either in spring or autumn.

## 3.2 Seasonal cycle

Significant cycles at different frequencies were detected via wavelet analysis at the BE Time-Series Station during 2005–2017 (Fig. 3). A half-year $NO_2^-$ cycle sporadically occurred in 2007–2009, 2013 and 2015. There is a seasonal $NO_2^-$ variability (at the frequency of 1 year) between 2007 and 2016 (times before 2007 and after 2016 were outside the conic line), except during 2010–2012, when high $NO_2^-$ concentrations were not observed in winter (Fig. 2). A biennial cycle of $NO_2^-$ could be observed as well during 2008–2015. The $NO_3^-$ concentrations were dominated by an annual cycle and a minor half-year cycle. The biennial cycle only occurred in 2008 and 2009. A remarkable seasonal variability of dissolved $O_2$ prevailed all the time, which is also obvious from the times series data shown in Fig. 2. The annual $N_2O$ cycle became gradually more and more evident until 2014, then declined and reoccurred less intensely in 2016. The periodical cycle was also present at other frequencies, indicated by the broadening of the red area before 2015 in Fig. 2d. For example, a biennial $N_2O$ cycle occurred during 2013–2015.

The half-year cycles of $NO_2^-$ and $NO_3^-$ were probably associated with algae blooms which usually occur in each spring and autumn (Smetacek, 1985; Bange et al., 2010). Since the time between the two blooms differed between years, the cycles were weak and thus not present in every year. Due to the fact that there was no half-year $O_2$ cycle at all, nutrients apart from $O_2$ might be the "drivers" of the sporadic half-year $N_2O$ cycle in 2008 and 2015, because $N_2O$ production depends on the concentration of the bioavailable nitrogen compounds (Codispoti et al., 2001).



Generally the wavelet analysis indicated a strong annual cycle for $NO_2^-$, $NO_3^-$, dissolved $O_2$ and
$N_2O$ at the BE Time-Series Station, which enabled us to explore the seasonal pattern with annual
mean data. Although extreme values were excluded as a result of averaging, the smoothed results
generally reflect the seasonality of these parameters. Here, we focus on the annual cycle.
The annual mean vertical distribution of dissolved $O_2$, $NO_2^-$, $NO_3^-$ and $N_2O$ are shown in Fig. 4.
Due to the development of stratification, the mixed layer was shallow in summer and deep in late
autumn/winter. $O_2$ depletion was observed in bottom waters from late spring until late autumn.
The seasonal distributions of $NO_2^-$ and $NO_3^-$ were similar and significantly correlated with high
concentrations observed in winter ($[NO_3^-]=11.59[NO_2^-]-0.51$, $R^2=0.80$, n=72, p<0.0001).
Minimum $N_2O$ concentrations were found in the bottom waters during September and October,
presumably as a result of consumption during denitrification under anoxic condition (Codispoti
et al., 2005). High $N_2O$ concentrations were observed in late spring and late autumn, respectively.
In late spring $N_2O$ accumulated in the bottom waters because the stratification prevented mixing
of the water column. In late autumn, however, $N_2O$ could be ventilated to the surface and thus
emitted to the atmosphere due to the breakdown of the stratification. The high $N_2O$
concentrations could be attributed to enhanced $N_2O$ production via nitrification and/or
denitrification within the oxic/anoxic interface (Goreau et al., 1980; Codispoti et al., 1992). Since
there is no clear $O_2$ concentration threshold, $N_2O$ production from both nitrification and the onset
of denitrification overlap at oxic/anoxic inteface. To this end, direct $N_2O$ production
measurements (i.e. nitrification/denitrification rates) are required to decipher which process
dominates the formation of the different $N_2O$ maxima.
High $N_2O$ concentrations prevailed all over the water column in winter/early spring. $NH_4^+$ is
released from the sediment into bottom waters due to the degradation of organic matter (Dale et
al., 2011), especially after the autumn algae bloom. The stratification usually completely breaks
down at this time of the year and the water column becomes oxygenated. Denitrification is
inhibited by the presence of $O_2$ and thus nitrification is presumably responsible for the high $N_2O$
concentrations in winter/early spring. This is supported by the positive correlation between pH
and $N_2O$ concentrations in December, January and February during 2005–2017 (Fig. 5). An
increase in pH would shift the $NH_3$:$NH_4^+$ equilibrium and provide more available $NH_3$ for
nitrification. Similar relationship between pH and $N_2O$ production also was reported in sub-polar
and polar Atlantic Ocean (Rees et al., 2016).

## 3.3 Trend analysis

The MKTs were conducted for the surface (1m) and bottom (25m) $N_2O$ concentrations and
saturations of the individual 12 months, respectively. Significant decreasing trends were detected
for the concentrations in the bottom waters for February and August (Table 1a), and for the
saturations in the surface for September and in the bottom for August and November (Table 1b).
These results indicated that some systematical changes in $N_2O$ took place at BE. For example,
the significant decrease in $N_2O$ concentration/saturation in August might be associated with the



increasing temperature, which reinforces the stratification and accelerates $O_2$ consumption in the
bottom waters (Lennartz et al., 2014). As a result, hypoxia/anoxia starts earlier and thus enables
the onset of denitrification to consume $N_2O$. During most of the months, trends in $N_2O$
concentration and saturation were not significant during 2005–2017.
A significant nutrient decline has been observed at the BE Time-Series Station since the mid-
1980s, however, Lennartz et al. (2014) found that bottom $O_2$ concentrations were still decreasing
over the past 60 years. The ongoing oxygen decline was attributed to the temperature-enhanced
$O_2$ consumption in the bottom water (Meier et al., 2018) and a prolongation of the stratification
period at the BE Time-Series Station (Lennartz et al., 2014). Please note that the trends in
nutrients and $O_2$ concentrations were detected based on the data collection which lasted for
approximately 30 and 60 years, respectively, while the $N_2O$ observations at BE Time-Series
Station has lasted for only 12.5 years. Further MKT analysis for nutrients, temperature and
oxygen for months with significant trends in $N_2O$ concentrations did not show any significant
results ($p>0.05$). The significant trends in $N_2O$ concentrations thus do not seem to be directly
related to one of these parameters, and we cannot state a reason for the significant trends of $N_2O$
concentration in February and the $N_2O$ saturation in September and November at this point.
Presumably, a longer monitoring period for $N_2O$ is required to detect corresponding trends in
$N_2O$ and oxygen or nutrients.

## 270   3.4 Extreme events

### 271   3.4.1 Low $N_2O$ concentrations during October 2016-April 2017

Besides the low $N_2O$ concentrations occurring in autumn, we observed a band of pronounced
low $N_2O$ concentrations which started in October 2016 and lasted until April 2017 (Fig. 6). In
this period $N_2O$ concentrations varied between 5.5–13.9 nM, with an average of 8.4±2.0 nM.
This is approximately 40% lower than the average $N_2O$ concentration during the entire
measurement period 2005–2017. The average $N_2O$ saturation during 2005–2017 was 111±30%,
while from October 2016 to April 2017, the $N_2O$ saturations were as low as 43–93% (average
278   62±10%).

Undersaturated $N_2O$ waters have been previously reported from the Baltic Sea: Rönner (1983)
observed a $N_2O$ surface saturation of 79% in the central Baltic Sea and attributed the
undersaturation to upwelling of $N_2O$-depleted waters. Bange et al. (1998) found a minimum $N_2O$
saturation of 91% in the southern Baltic Sea where the hydrographic conditions were
significantly influenced by riverine runoff. Walter et al. (2006) reported a mean $N_2O$ saturation
of 79±11% for shallow stations (<30 m) in the southwestern Baltic Sea in October 2003. The
low-$N_2O$ event at BE was unusual because the concentrations were much lower than those
reported values and it lasted for more than half a year.



Although the observed temperatures and salinities during October 2016–April 2017 were
comparable to other years, it is difficult to evaluate the role of physical mechanism in the low-
$N_2O$ event because of insufficient data for water mass exchange at the BE Time-Series Station.
Here we mainly focused on the chemical or biological processes. Anoxia events with the
presence of $H_2S$ were observed in the bottom waters for three months in a row during
September–November 2016. This is an unusual long period and is unprecedented at the BE
Time-Series Station. In December 2016 the stratification did not completely break down.
Although the water column was generally oxygenated, bottom $O_2$ concentrations were the lowest
observed during the past ten years. Considering the classical view of $N_2O$ consumption via
denitrification under hypoxic and anoxic conditions, we inferred that denitrification accounted
for low $N_2O$ concentrations in the bottom layer. However, the question still remains where the
low-$N_2O$-concentration water in the upper layers came from.
In September 2016, low $N_2O$ concentrations were only observed in the bottom waters where the
anoxia occurred. However, the situation was different in the following months. During
October/November 2016, $N_2O$ concentrations were homogeneously distributed in the water
column. Although the stratification gradually started to break down in late autumn, the density
gradient was still strong enough to keep the bottom waters at anoxic conditions and prevented
the low-$N_2O$-concentration to reach the surface. Thus we inferred that the unusual low $N_2O$
concentrations in the upper layers (above 20 m) were probably resulting from advection of
adjacent waters. Due to the fact that the upper layers were well-mixed and oxygenated, in situ
$N_2O$ consumption in the water column could be neglected. We suggest therefore, that the $N_2O$
depleted waters were resulting from consumption of $N_2O$ in bottom waters elsewhere and then
they were upwelled and transported to BE. Hence, $N_2O$ consumption via denitrification might
have been, directly or indirectly, responsible for the low $N_2O$ concentrations during October–
November 2016.
In December 2016, the bottom waters were ventilated with $O_2$. Although $N_2O$ consumption by
denitrification should have been inhibited by the presence of $O_2$ (Codispoti et al., 2001), the $N_2O$
concentrations did not restore to their normal level under suboxic conditions. Since January 2017,
the whole water column was well mixed and oxygenated. Usually a significant nutrient supply
could be observed starting in November (Fig. 4) as a result of remineralization and vertical
mixing, but the average $NO_2^-$ and $NO_3^-$ concentrations during November 2016–April 2017 were
0.2 and 1.4 µM, respectively, which was about 50% and 60% lower than in other years.
Ammonium ($NH_4^+$) and Chlorophyll *a* concentrations (data not shown) during this period were
comparable to that of other years. We did not observe an exceptional spring algae bloom in 2017,
indicating that assimilative uptake by phytoplankton was not responsible for the low nutrients
concentrations. The nutrient deficiency might be attributed to enhanced nitrogen removal
processes like denitrification or anammox (Voss et al., 2005; Hietanen et al., 2007; Hannig et al.,
2007) during the prolonged period of anoxia in autumn 2016. During the low $N_2O$ event, we
found    that    $N_2O$    concentrations    were    positively    correlated    with    both    $NO_2^-$



($[N_2O]=7.02[NO_2^-]+7.36$, $R^2=0.29$, n=24, p<0.01) and $NO_3^-$ ($[N_2O]=0.80[NO_3^-]+7.36$, $R^2=0.51$,
n=24, p<0.0001). These results indicate that the development and maintenance of the low-$N_2O$-
concentration was closely associated with nutrient deficiency. Especially after the breakdown of
the stratification, when denitrification was no longer a significant $N_2O$ sink, nutrients might have
become a limiting factor for $N_2O$ production.
In general, the low-$N_2O$-concentration event during October 2016–April 2017 can be divided
into two parts: in the stratified waters during October–November 2016, $O_2$ played a dominant
role and $N_2O$ was consumed via denitrification under anoxic conditions. In the well-mixed water
column during December 2016–April 2017, nutrient deficiency seemed to have constrained $N_2O$
production via nitrification under suboxic/oxic conditions.
In recent years a novel biological $N_2O$ consumption pathway, called $N_2O$ fixation, which
transforms $N_2O$ into particulate organic nitrogen via its assimilation, has been reported (Farías et
al., 2013). This process can take place under extreme environmental conditions even at very low
$N_2O$ concentrations. Cornejo et al. (2015) reported that $N_2O$ fixation might play a major role in
the coastal zone off central Chile where seasonally occurring surface $N_2O$ undersaturation was
observed. The relatively high $N_2$ fixation rates in the Baltic Sea (Sohm et al., 2011) highlight the
potential role of $N_2O$ fixation (Farías et al., 2013). However, we cannot quantify the role of
biological $N_2O$ fixation for the $N_2O$ depletion in the Baltic Sea due to the absence of $N_2O$
assimilation measurements.

### 345    3.4.2 High $N_2O$ concentrations in November 2017

High $N_2O$ concentrations were observed at the BE Time-Series Station in November 2017. The
average value reached 35.4±1.5 nM, which was the highest concentration measured during the
entire sampling period from 2005 to 2017. Dissolved $N_2O$ was homogeneously distributed in the
water column, but this event did not last long. In December, dissolved $N_2O$ returned to normal
levels and the average concentration in the water column was comparable to that of other years.
Average $N_2O$ saturation in November 2017 was 322±10%, which was also the highest for the
past 12.5 years. This value was much higher than the maximum surface $N_2O$ saturation reported
by Rönner (1983) in the central Baltic Sea, but was comparable to the results observed in the
southern Baltic Sea (312%, Bange et al., 1998). Bange et al. (1998) linked the enhanced $N_2O$
concentrations to riverine runoff because those samples were collected in an estuarine area,
however, the riverine influence around the BE Time-Series Station is negligible. As a result, the
impact of fresh water input can be excluded.
Dissolved $O_2$ seemed to play a dominant role in the high $N_2O$ concentrations. Enhanced $N_2O$
production usually occurred at the oxic/anoxic interface, which was closely linked to the
development of water column stratification. In general the breakdown of the stratification is
faster than its establishment at the BE Time-Series Station. As a result, it took about half a year
for bottom $O_2$ saturation to gradually decrease from ~80% to almost 0% (i.e. anoxia), but only



two months to restore normal saturation level in 2010 (Fig. 7). In late autumn, surface water penetrated into the deep layers via vertical mixing and eroded the oxic/anoxic interface. The entire water column quickly became oxygenated and the enhanced $N_2O$ production was stopped.

Hypoxia/anoxia at BE is usually observed in the bottom waters in autumn, but in September 2017, hypoxic water ($O_2$ saturation<20 %, which was close to the criterion for hypoxia, see Naqvi et al., 2010) was found in the subsurface layer (10 m) as well. Surface $O_2$ saturation was only ~50%, which was the lowest during the sampling period 2005–2017. The density gradient of the water column in September 2017 was much lower than in other years. These results indicate the occurrence of an upwelling event at BE Time-Series Station in autumn 2017, which might be a result of the saline water inflow from the North Sea considering the change of salinity in the water column. Strong vertical mixing has interrupted the hypoxia/anoxia and bottom $O_2$ saturation reached ~60% in October 2017. The presence of $O_2$ prevented $N_2O$ consumption via denitrification, as a result, we did not observe a significant $N_2O$ decline during that period (Fig. 6).

Considering the fact that a significant autumn algae bloom was observed in autumn 2017 (as indicated by high chlorophyll a concentrations, data not shown), severe $O_2$ depletion in the bottom water could be expected. Although the bottom $O_2$ saturation was only slightly lower in November than in October, we speculate that even lower $O_2$ saturation (but not anoxia) might have occurred between October and November. The "W-shaped" $O_2$ saturation curve (see Fig. 7) suggests that the stratification did not completely break down in October and that there might have been a reestablishment of the oxic/anoxic interface providing favorable conditions for enhanced $N_2O$ production. Due to the degradation of organic nitrogen, $NH_4^+$ is released from the sediment into bottom waters (Dale et al., 2011), especially in autumn when $O_2$ is low. $NH_4^+$ concentrations in November 2017 were lower than in other years, and $NO_2^-$ concentrations were higher, indicating that nitrification occurred in bottom waters. To this end, we suggest that the reestablishment of the oxic/anoxic interface promoted ammonium oxidation (the first step of nitrification). In this case, $N_2O$ could have temporary accumulated because its consumption via denitrification was blocked. Meanwhile, the relatively low density gradient (i.e. low stratification) allowed upward mixing of the excess $N_2O$ to the surface. However, we inferred that that this phenomenon would only last for a few days due to the rapid breakdown of stratification at the BE Time-Series Station.

Due to the development of the pronounced stratification, the oxic/anoxic interface prevailed in summer/early autumn as well, but we did not observe $N_2O$ accumulation during these months. One of the potential explanations is that enhanced $N_2O$ production only took place within particular depths where strong $O_2$ gradient existed, but our vertical sampling resolution was too low to capture this event. Also enhanced $N_2O$ production might be covered by the weak mixing which brought low-$N_2O$ water from the bottom to the surface.



The upwelling event played different roles in autumn 2016 and 2017. First, upwelling took place
somewhere else but at BE because of the strong density and $O_2$ gradient in the water column
during autumn 2016. Second, bottom water remained anoxic in autumn 2016, while the
compensated water for upwelling in 2017 penetrated through stratification and brought $O_2$ into
bottom water (Fig. 7), which caused enhanced $N_2O$ production. Similarly, autumn upwelling was
detected in 2011 and 2012 when we found relatively low $O_2$ concentrations in subsurface layers
(10 m) (Fig. 2), but we did not observe an increase in bottom $O_2$ concentrations and $N_2O$
concentrations remained low during that time. These upwelling events seem to be driven by
saline water inflow considering the prominent increase in salinity, but the mechanism dominates
$O_2$ input into bottom water before the stratification break down remains unclear.

## 3.5 Flux density

During 2005–2017, surface $N_2O$ saturations at the BE Time-Series Station varied from 56 % to
314 % (69–194 % excluding the extreme values discussed in Sect. 3.4), with an average of
111±30 % (111±20 % without the extreme values). Generally the water column at BE was
slightly oversaturated with $N_2O$. Our results are in good agreement with the estimated mean
surface $N_2O$ saturation for the European shelf (113%, Bange, 2006).
We found a weak seasonal cycle for surface $N_2O$ concentrations, with high $N_2O$ concentrations
occurring in winter/early spring and low concentrations occurring in summer/autumn (Fig. 4; Fig.
8a, b). No pronounced seasonality was found for $N_2O$ saturation. The trend in concentration but
not in saturation likely results from the modulating effect of temperature on $N_2O$ emissions:
During wintertime when temperatures are low, the solubility of $N_2O$ increases, which lowers
emissions and increases the $N_2O$ concentration in the surface water. The calculation of $N_2O$
saturation takes this temperature effect into account and, hence, does not show any pronounced
seasonality. We thus conclude that temperature plays an important modulating role for $N_2O$
emissions.
The wind speed ($u_{10}$) at the BE Time-Series Station ranged from 1.1 to 14.0 m s$^{-1}$, with an
average of 7.0±2.7 m s$^{-1}$. $N_2O$ flux densities varied from -19.0 to 105.7 µmol m$^{-2}$ d$^{-1}$ (-14.1–30.3
µmol m$^{-2}$ d$^{-1}$ without the extreme values), with an average of 3.5±12.4 µmol m$^{-2}$ d$^{-1}$ (3.3±6.5
µmol m$^{-2}$ d$^{-1}$ without the extreme values). However, the true emissions might have been
underestimated because our monthly sampling resolution is insufficient to capture short-term
$N_2O$ accumulation events due to the fast breakdown of stratification in autumn. Our flux
densities are comparable to those reported by Bange et al. (1998, 0.4–7.1 µmol m$^{-2}$ d$^{-1}$) from the
coastal waters of the southern Baltic Sea, but are slightly lower than the average $N_2O$ flux
density reported by Rönner (1983, 8.9 µmol m$^{-2}$ d$^{-1}$) from the central Baltic Sea. Please note that
the results of Rönner (1983) were obtained only from the summer season and therefore are
probably biased because of missing seasonality.

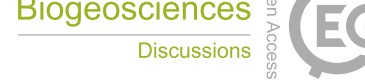



In December 2014, a strong saline water inflow from the North Sea was observed, which was the third strongest ever recorded (Mohrholz et al., 2015). Although the salinity in December 2014 was comparable to other years, a remarkable increase in salinity was observed in the following several months. However, we did not detect a significant $N_2O$ anomaly or enhanced emission during that time. Similarly, Walter et al. (2006) investigated the impact of the North Sea water inflow on $N_2O$ production in the southern and central Baltic Sea in 2003. The oxygenated water ventilated the deep Baltic Sea and shifted anoxic to oxic condition which led to enhanced $N_2O$ production, but the accumulated $N_2O$ was unlikely to reach the surface due to the presence of a permanent halocline (Walter et al., 2006).

Although we observed extremely high $N_2O$ flux density in November 2017, the low-$N_2O$-concentration (<10 nM) events have become more and more frequent during the past ten years (Fig. 2). This phenomenon seldom occurred before 2011, but remarkable low $N_2O$ concentrations can be seen in 2011 and 2013, and to a less extent in 2012 and 2014. Similar events lasted for several months in 2015 and for even more than half a year during 2016–2017. The most striking was that the low-$N_2O$-concentration water was not only detected in bottom waters, but also at surface which would significantly impact the air-sea $N_2O$ flux densities. Although the MKT result did not give a significant trend for the $N_2O$ flux densities, the data presented in Fig. 9 suggest a potential decline of $N_2O$ flux densities from the coastal Baltic Sea, challenging the conventional view that $N_2O$ emissions from coastal waters would most probably increase in the future, which was based on the hypothesis of increasing nutrient loads into coastal waters. Due to an effective reduction of nutrient inputs, the severe eutrophication condition in the Baltic Sea has been alleviated (HELCOM, 2018b), but ongoing deoxygenation points to the fact that it will take a longer time for coastal ecosystems to feedback to reduced nutrient inputs because other environmental changes such as warming may override decreasing eutrophication (Lennartz et al., 2014).

## 4. Conclusions

The seasonal and inter-annual $N_2O$ variations at the BE Time-Series Station from July 2015 to December 2017 were driven by the prevailing $O_2$ regime and nutrients availability. We found a pronounced seasonal cycle with low $N_2O$ concentrations (undersaturations) occurring in hypoxic/anoxic bottom waters in autumn and enhanced concentrations (supersaturations) all over the water column in winter/early spring. Significant decreasing trends for $N_2O$ concentrations were found for few months, while most of the year, no significant trend was detectable in the period of 2005–2017. During 2005–2017, no significant trends were present for $O_2$ and nutrients either, but these parameters all show significant decreasing trends on longer time scales (~60 years) at BE. Our results show the strong coupling of $N_2O$ with $O_2$ and nutrient concentrations, and suggest similar changes on comparable time scales. Further monitoring of $N_2O$ at BE time series station is thus important to detect changes. Further studies on $N_2O$ production/consumption by nitrification and denitrification and analysis of the characteristic $N_2O$




isotope signature might be very helpful to decipher the potential roles of $O_2$/nutrients for $N_2O$
cycling.
Temperature plays a modulating role for the $N_2O$ emission at the BE Time-Series Station.
Although the hydrographic condition at BE is generally dominated by the inflow of saline North
Sea water, this did not affect $N_2O$ production and its emissions to the atmosphere. It seems that
events with extremely low $N_2O$ concentrations and thus reduced $N_2O$ emissions became more
frequent in recent years. Our results provide a new perspective onto potential future patterns of
$N_2O$ distribution and emissions in coastal areas. Continuous measurement at the BE Time-Series
Station with a focus on late autumn would be of great importance for monitoring and
understanding the future changes of $N_2O$ concentrations and emissions in the southwestern Baltic
Sea.

## Author contribution

X.M., S.T.L. and H.W.B. designed the study and participated in the fieldwork. $N_2O$
measurements and data processing were done by X.M. and S.T.L. X.M. wrote the manuscript
with contributions from S.T.L. and H.W.B.

## Acknowledgments

The authors thank the captain and crew of the RV *Littorina* and *Polarfuchs* as well as the many
colleagues and numerous students who helped with the sampling and measurements of the BE
time-series through various projects. Special thanks to A. Kock for her help with sampling,
measurements and data analysis. The time-series at BE was supported by DWK
Meeresforschung (1957–1975), HELCOM (1979–1995), BMBF (1995–1999), the Institut für
Meereskunde (1999–2003), IfM-GEOMAR (2004–2011) and GEOMAR (2012–present). The
current $N_2O$ measurements at BE are supported by the EU BONUS INTEGRAL project which
receives funding from BONUS (Art 185), funded jointly by the EU, the German Federal
Ministry of Education and Research, the Swedish Research Council Formas, the Academy of
Finland, the Polish National Centre for Research and Development, and the Estonian Research
Council. The Boknis Eck Time-Series Station (www.bokniseck.de) is run by the Chemical
Oceanography Research Unit of GEOMAR, Helmholtz Centre for Ocean Research Kiel. Data
from BE are available from www.bokniseck.de/database-access. The $N_2O$ data presented here
have been archived in MEMENTO (the MarinE MethanE and NiTrous Oxide database:
https://memento.geomar.de). X. Ma is grateful to the China Scholarship Council for providing
financial support (File No. 201306330056) and the EU BONUS INTEGRAL project.

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





Table 1. The results of the Mann-Kendall test for the surface and bottom $N_2O$ concentrations and
saturations of the 12 individual months.

684                                   Table 1a. MKT results for $N_2O$ concentrations

| Month | January | | February | | March | | April | |
|---|---|---|---|---|---|---|---|---|
| Depth/m | 1 | 25 | 1 | 25 | 1 | 25 | 1 | 25 |
| p | 0.09 | 0.19 | 0.11 | 0.03(-) | 0.19 | 0.63 | 0.09 | 0.30 |
| Month | May | | June | | July | | August | |
| Depth/m | 1 | 25 | 1 | 25 | 1 | 25 | 1 | 25 |
| p | 0.63 | 0.24 | 0.15 | 0.95 | 0.16 | 0.16 | 0.20 | 0.03(-) |
| Month | September | | October | | November | | December | |
| Depth/m | 1 | 25 | 1 | 25 | 1 | 25 | 1 | 25 |
| p | 0.25 | 0.76 | 0.36 | 0.76 | 0.67 | 0.16 | 0.10 | 0.30 |


686                                   Table 1b. MKT results for $N_2O$ saturations

| Month | January | | February | | March | | April | |
|---|---|---|---|---|---|---|---|---|
| Depth/m | 1 | 25 | 1 | 25 | 1 | 25 | 1 | 25 |
| p | 0.37 | 0.24 | 0.15 | 0.15 | 0.19 | 0.63 | 0.11 | 0.19 |
| Month | May | | June | | July | | August | |
| Depth/m | 1 | 25 | 1 | 25 | 1 | 25 | 1 | 25 |
| p | 0.19 | 1 | 0.37 | 0.54 | 0.10 | 0.43 | 0.20 | 0.02(-) |
| Month | September | | October | | November | | December | |
| Depth/m | 1 | 25 | 1 | 25 | 1 | 25 | 1 | 25 |
| p | 0.04(-) | 0.85 | 0.06 | 0.43 | 0.20 | 0.03(-) | 0.16 | 0.36 |


p indicates the p-value of the test, which is the probability, under the null hypothesis, of obtaining a value of
the test statistic as extreme or more extreme than the value computed from the sample.
(-) indicates a rejection of the null hypothesis at α significance level and a decreasing trend is detected.





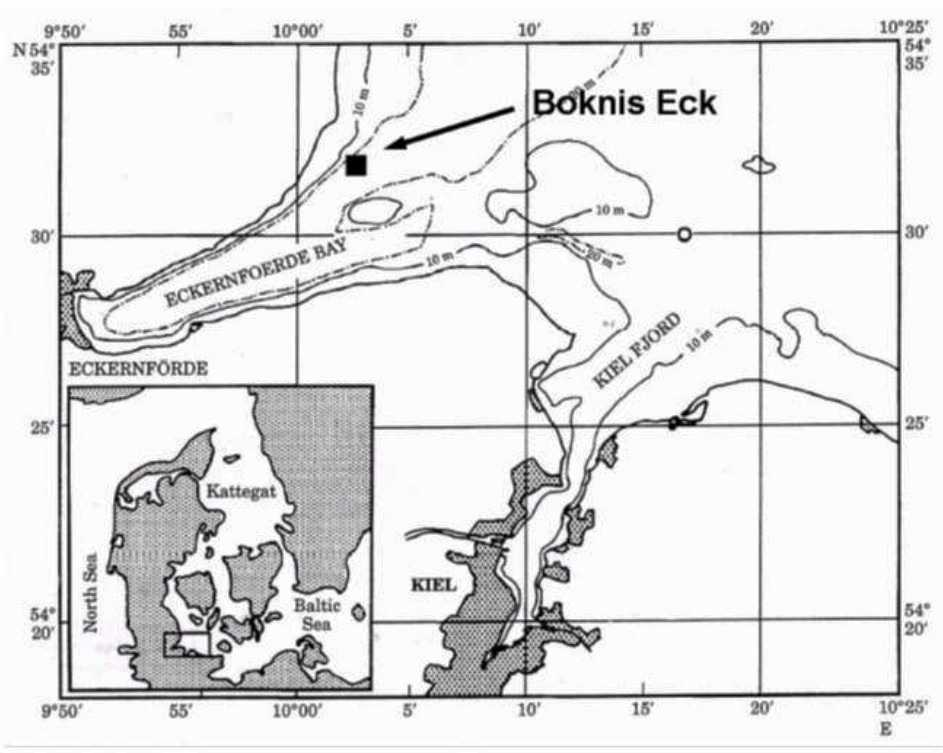


Fig. 1 Location of the Boknis Eck Time-Series Station in the Eckernförde Bay, southwestern Baltic Sea. (Map
from Hansen et al., 1999)





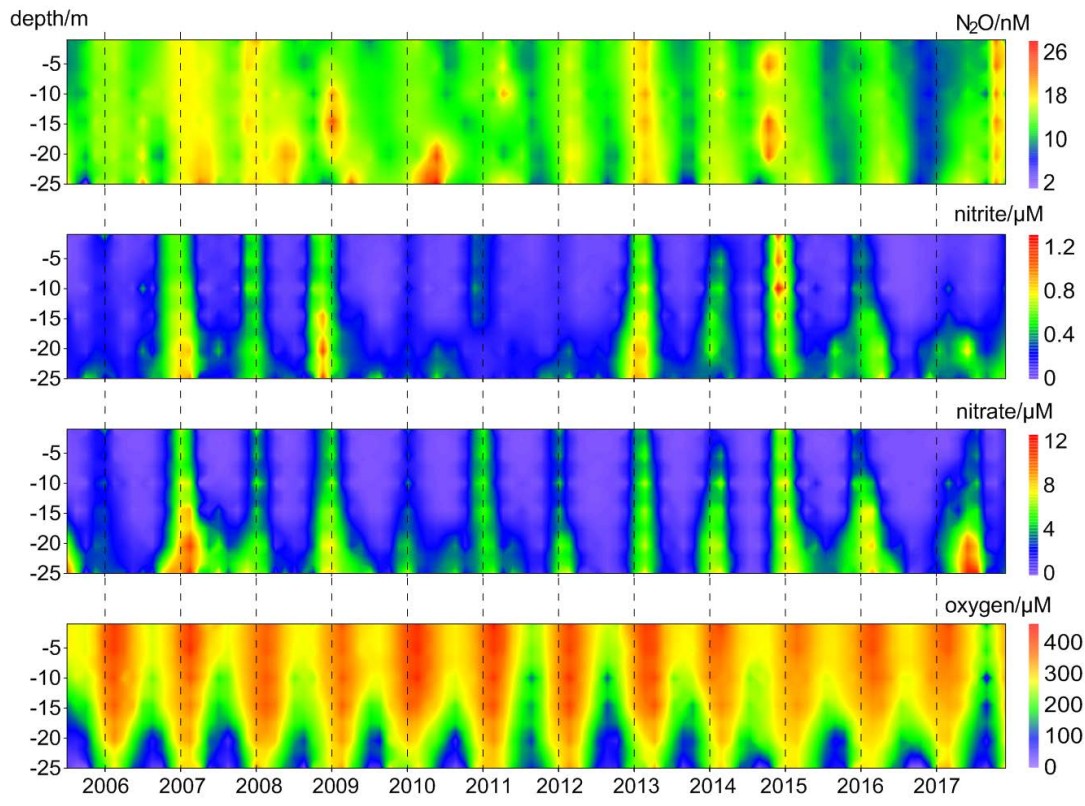

Fig. 2 Vertical distributions of dissolved $O_2$, $NO_2^-$, $NO_3^-$, and $N_2O$ from the BE Time-Series Station during 2005–2017.





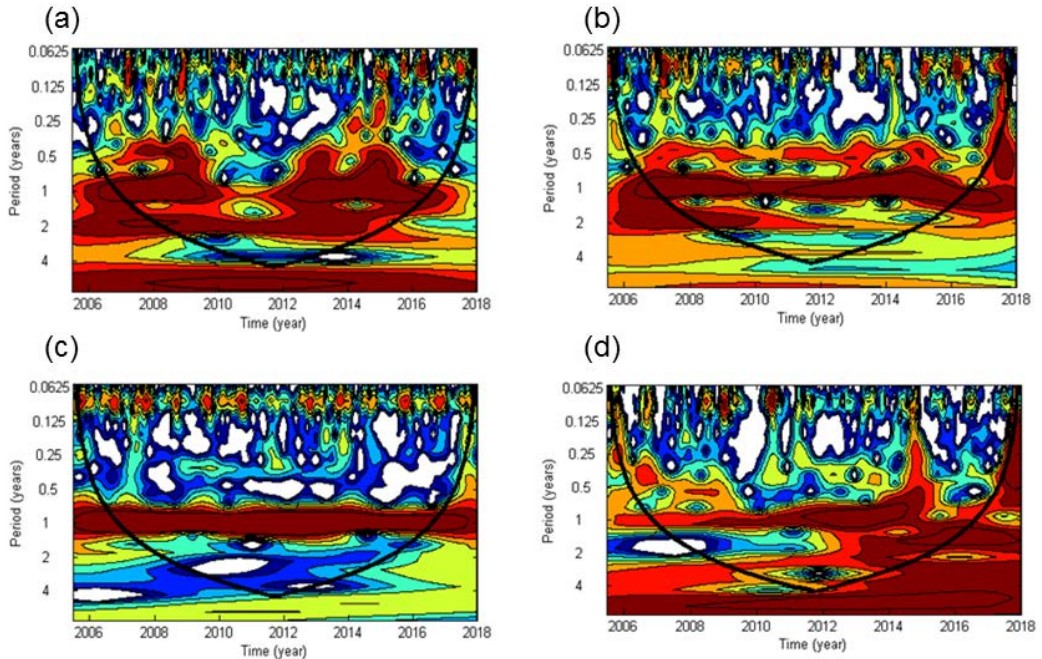


Fig. 3 Wavelet power spectra of $NO_2^-$ (a), $NO_3^-$ (b), dissolved $O_2$ (c) and $N_2O$ (d) from the BE Time-Series
Station. Red areas indicate high, blue indicate low power. The black conic line indicates the significant area
where boundary effects can be excluded.




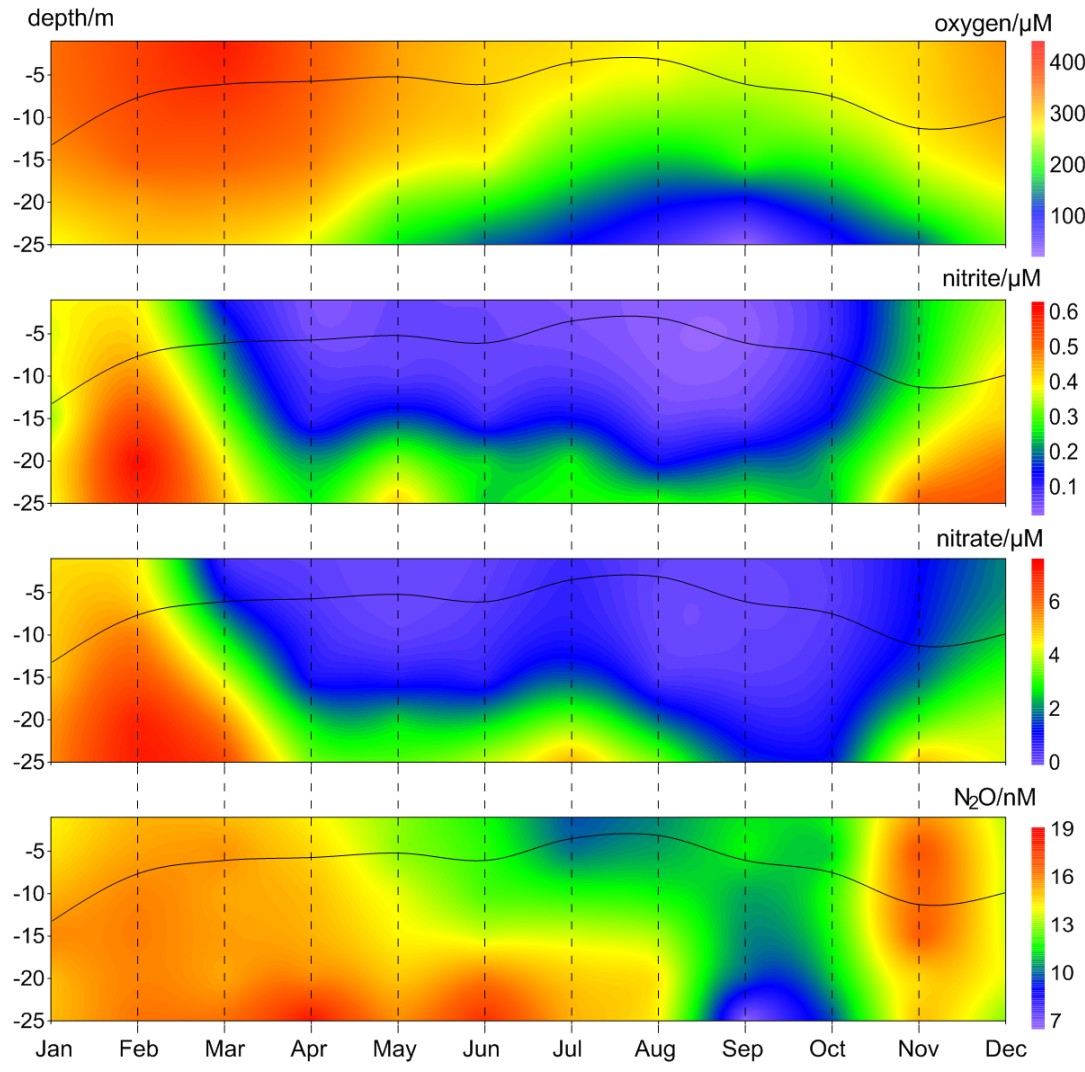

Fig. 4 Average vertical distributions of dissolved $O_2$, $NO_2^-$, $NO_3^-$, and $N_2O$ from the BE Time-Series Station during 2005–2017. The black line indicates the mixed layer depth, which was calculated based on a potential density anomaly of 0.15 kg m$^{-3}$ from the sea surface (1m).



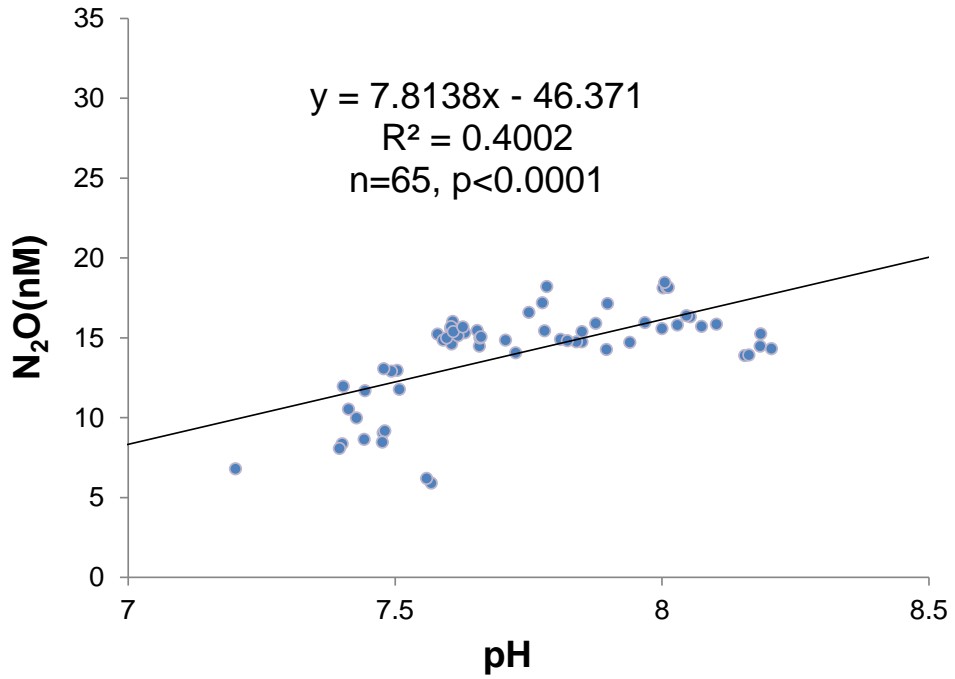

705
706    Fig. 5 Correlation between pH and $N_2O$ concentrations in December, January and February during 2005–2017.





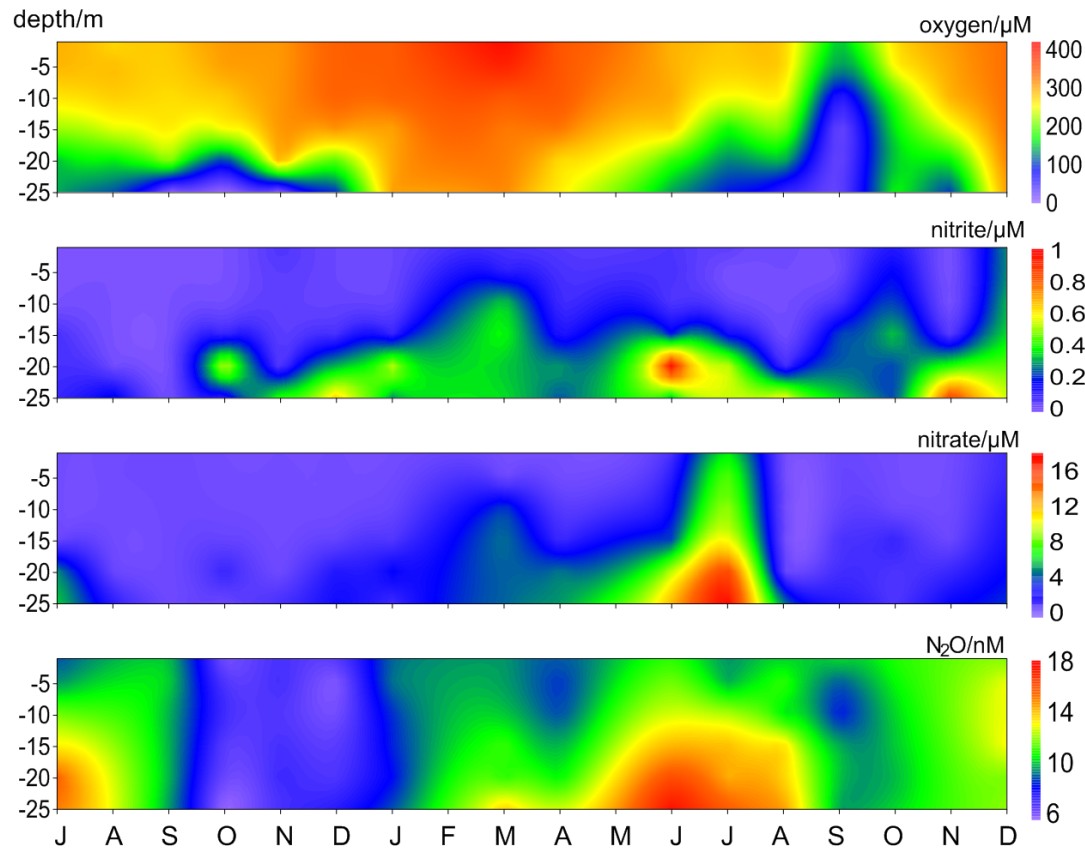

Fig. 6 Vertical distribution of dissolved $O_2$, $NO_2^-$, $NO_3^-$, and $N_2O$ from the BE Time-Series Station during July 2016–December 2017. Please note that the high $N_2O$ concentrations in November 2017 were removed for better visualization.





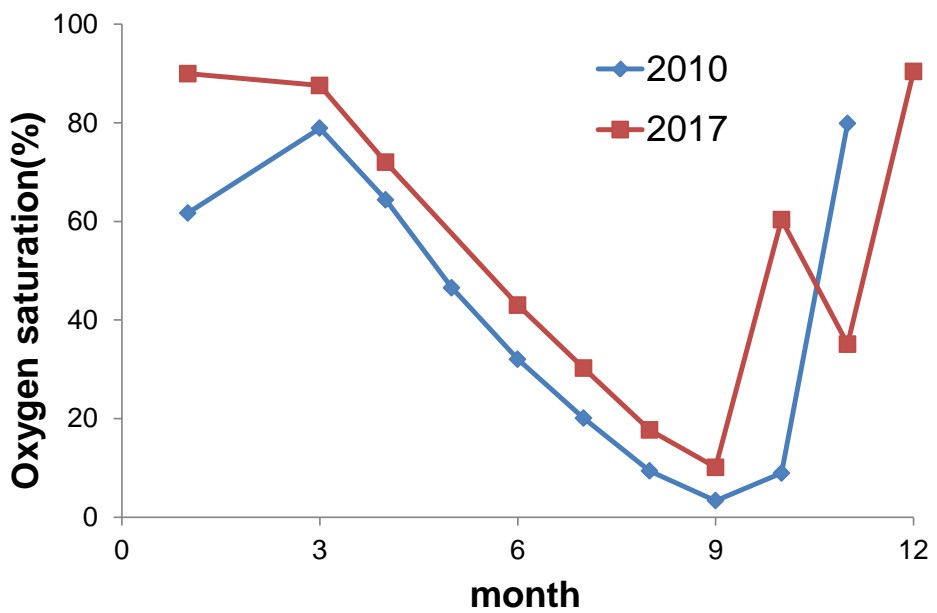


Fig. 7 Variations of bottom $O_2$ saturation in 2010 (blue) and 2017 (red).






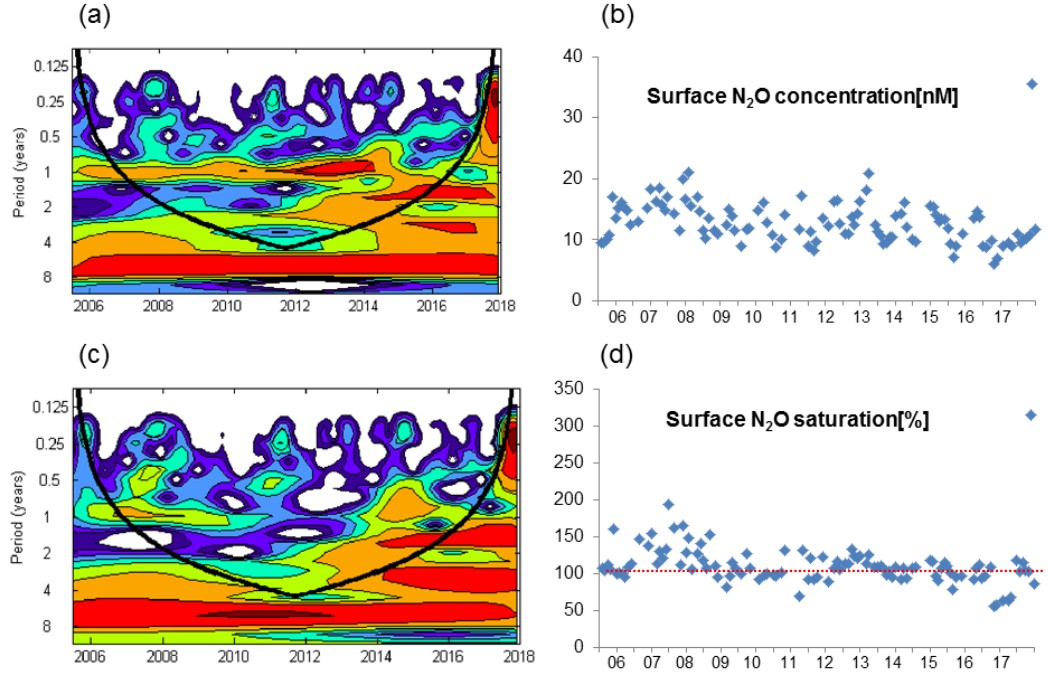

Fig. 8 Wavelet analysis and the variation of surface N$_2$O concentrations (a, b) and surface N$_2$O saturations (c,
d). The dashed red line in (d) indicates the saturation of 100%.





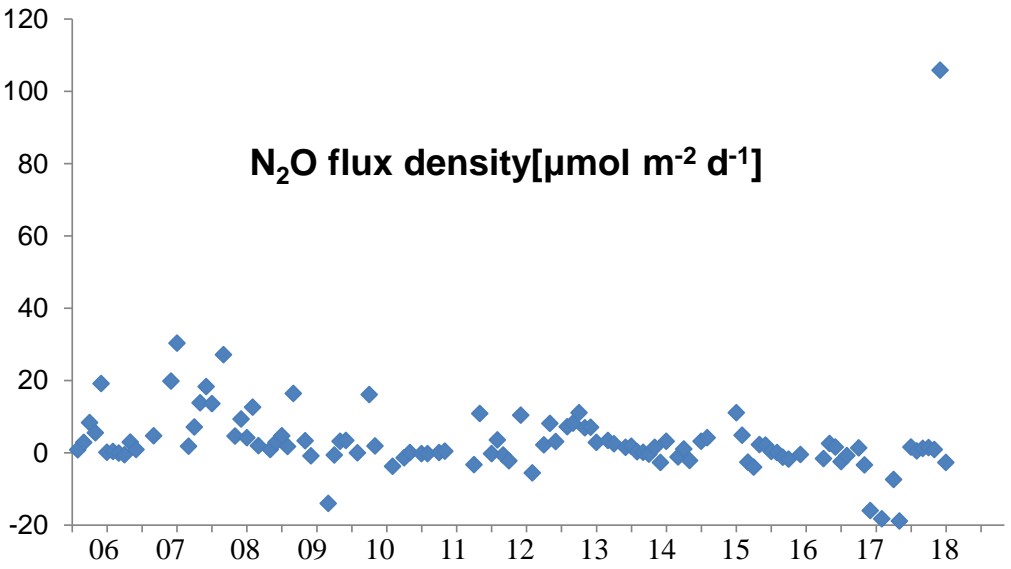


Fig. 9 Variation of $N_2O$ flux density at the BE Time Series-Station during 2005–2017. Negative values
indicated $N_2O$ influx from the atmosphere and positive values indicated $N_2O$ efflux to the atmosphere.