# Peer review of "A multi-year observation of nitrous oxide at the Boknis Eck"

_Biogeosciences, 2019_

## Referee Comment (RC1) · Anonymous Referee #1 · 4 Jul 2019

General comments:

The present paper examined the seasonal and annual variations of dissolved N2O in a time-series station located in the southwestern Baltic Sea. The results show the coupled variations between the N2O anomalies, the oxygen concentrations, and nutrients.

The paper presents a valuable new dataset of N2O and related biogeochemical parameters in a marine region subject to extensive human activities and so nutrients inputs, responsible of the deoxygenation in the Baltic sea. After the revision, I consider that the manuscript is highly interesting and provide relevant information about processes occurring to the N2O in the Boknis Eck. The paper is well written and structured, with

an appropriate description of the state of the art, objectives are clearly outlined and discussion precisely referenced. The main strength of the paper is the monthly sampling undertaken during twelve years.

However, there are several weaknesses in the paper. First, the authors make the discussion of the results based on data that are included in this study. There are references to data that exist but are not shown. But, in other to discuss about upwelling and hydrographic changes, about algal blooms and ammonium changes, the salinity, temperature, chlorophyll and ammonium data should be included and shown in this paper. Secondly, the paper lack of a proper description of the water masses presents at BE and their temporal variability.

Specific comments:

-Lines 129-130: How did you shifted the data to the 15th? Include procedure and assumptions in the text. Line 170: Could you explicitly explain in the text how did you computed Sc, instead just give the reference? what is the equation for computing Sc?

Lines 176-184: The comparison of the range of concentrations found between Boknis and other time-series would be better move later in the text, since the reader at this point does not have enough information about the causes that differentiate it from other time-series. The authors should better discuss not only the different magnitudes of the N2O concentrations, but also the site-specific processes responsible of such differences.

Lines 207...: In case there is additional information in the BTS, such as chlorophyll, during the study period, show the data in figures instead to refer to previous studies.

Lines 235: are there NH4 data available at the study site during the study period? In that case, it would be better to show them for the discussion instead to appeal to a reference

Lines 238-239: "Denitrification is inhibited by the presence of O2 and thus nitrification

is presumably responsible for the high N2O concentrations in winter/early spring." This statemen is not correct at all. The production of N2O by denitrification can occur at suboxic and hypoxic environments. Please, modify this sentence.

Line 239-240: The authors should normalized the N2O and pH to a constant temperature. Otherwise, temperature changes can be the responsible of this relationship because of thermodynamics changes and not necessarily due to nitrification. In fact, it is not as clear the positive correlation between the N2O and pH in figure 5, since for pH higher than 7.6, there is no apparent trend between N2O and pH. The relationship between pH and N2O obtained during incubations experiments described by Rees et al (2016) cannot be directly compare to this study, since the experimental conditions and approaches are completely different. The authors should rewrite the entire paragraph.

Lines 263-269: Again, the temperature salinity and Chla information at Boknis are mentioned in the text, but data are missed. If data for these parameters exist, the authors should include in the manuscript. It would reinforce some of the statement that now could look only speculative.

Lines 287-288: "Although the observed temperatures and salinities during October 2016–April 2017 were comparable to other years,..". Please, show temperature and salinity. Lines 295-296: "Considering the classical view of N2O consumption via denitrification under hypoxic and anoxic conditions". This is contrary to the statement done at lines 238-239. Consider to rewrite the first one. Lines 304-306; 308-309: The authors should make use of temperature, salinity or density to show changes in water masses. Lines 313: Instead of "presence" it would be more correct "concentration/level" Lines 320: "We did not observe an exceptional spring algae bloom in 2017". Please, consider to include Chla or POM to support this statement. Lines 319: Why can not be shown the Chla data?

Lines 331-335: The author should also discuss the potential dependence of rates on temperature and its impact on the seasonal variations of N2O production/consumption

trough the text.

Lines 356-357: Please, consider to support this statement with the salinity data 371-373. Please, show the density (or temperature and salinity) record to track the up-welling event in autumn 2017. Lines 377-378: Please, show the chlorophyll data. Lines 385_386: Please, show the ammonium data. Lines 394-399: This is a very speculative paragraph as it is written. Could you give any evidence for these potential explanations of the homogeneous distribution of N2O?

-Section 3.5: The author should evaluate in the results the impact of the dissolved gas analysis uncertainty in the air-sea flux computation and the uncertainty introduced in the net seasonal and annual air-sea NO fluxes.

Lines 416-424: The authors show that N2O concentration change seasonally, but the saturation stay almost constant. So, how can the author affirm that emissions are controlled by temperature?

Lines 476-484: Unless the author do not include salinity and temperature, they should not used them to conclude the hydrographic conditions at Boknis Eck. Further studies about the hydrography at the BE would complete the picture together with the biogeo-chemical data at the BE time-series station.

---

## Referee Comment (RC2) · Anonymous Referee #2 · 30 Jul 2019

Quantifying the concentrations and dynamics of dissolved N2O in seawater is important for understanding the climate change, but conducting measurements of sufficient duration to determine trends over seasonal, interannual, and decadal time frames for any marine ecosystem remains a challenging task. A long-term Time-Series Station like Boknis Eck in the Eckernförde Bay can provide invaluable information for documenting the role of oceans in relation to N2O, hence this type of study is significant for our scientific understanding. The paper is well-written and clear. I only have few comments/suggestions below: Lines 95-96: more information is needed on seawater sample collection. Lines 108-109: The N2O concentration of standard gases should be provided. Line 154-155: . . .dry mole fractions of atmospheric N2O at the time of

the sampling. This description is not the fact since atmospheric N2O at the time of the sampling has not been measured and the monthly average of N2O data measured at Mace Head was used to N2Oeq. Lines 185-193: NH4+ concentrations should be provided in the text as well as Figure 2. Lines 221-222: The expression caused misunderstanding. Lines 239-243: Does N2O correlate with NH4+? Line 462: The year of 2015 should be 2005? Line 476: There is no temperature data provided at all in this study but with a conclusion 'Temperature plays a modulating role for the N2O emission at the BE Time-Series Station'. I suggest to provide t data in Figure 2 and more t data provided in related discussion in the text Figure 2: It would be better for the authors to provide the vertical profiles of t, s, density, NH4+ and Chl a. Figure 6: The vertical profiles of hydrological parameters, such as t, s and density are needed to help understand the possible influence of physical processes on N2O distribution as discussed between lines 301 and 309. Figure 8: title is needed for x- and y-axis at figure b and d Figure 9: title is needed for x- and y-axis

---

## Author Comment (AC1) · 19 Aug 2019

Thank you very much for the comments. They are thought-provoking and helpful to improve our manuscript.

General comments: The present paper examined the seasonal and annual variations of dissolved N2O in a time-series station located in the southwestern Baltic Sea. The results show the coupled variations between the N2O anomalies, the oxygen concentrations, and nutrients. The paper presents a valuable new dataset of N2O and related biogeochemical parameters in a marine region subject to extensive human activities and so nutrients inputs, responsible of the deoxygenation in the Baltic Sea. After the

revision, I consider that the manuscript is highly interesting and provide relevant information about processes occurring to the N2O in the Boknis Eck. The paper is well written and structured, with an appropriate description of the state of the art, objectives are clearly outlined and discussion precisely referenced. The main strength of the paper is the monthly sampling undertaken during twelve years. However, there are several weaknesses in the paper. First, the authors make the discussion of the results based on data that are included in this study. There are references to data that exist but are not shown. But, in other to discuss about upwelling and hydrographic changes, about algal blooms and ammonium changes, the salinity, temperature, chlorophyll and ammonium data should be included and shown in this paper.

Reply: Both reviewers requested to show additional temperature, salinity, chlorophyll a and ammonium data. These data were not explicitly shown in our ms because they have been published already in Lennartz et al.: Long-term trends at the Boknis Eck time series station (Baltic Sea), 1957–2013: does climate change counteract the decline in eutrophication? Biogeosciences, 11, 6323–6339, https://doi.org/10.5194/bg-11-6323-2014.). However, in order to provide this obvious lacking information, we decided to include the seasonal and annual variations of temperature, salinity, chlorophyll a and ammonium in a new supplement to our manuscript.

Secondly, the paper lack of a proper description of the water masses presents at BE and their temporal variability.

Reply: Considering that all the measurements were conducted at one fixed location, it is difficult to investigate the water masses based on our data. However, the hydrological condition at BE are not complicated. As is written in Lines 86-87, there is no pronounced river input, and saline water from the North Sea plays a dominant role. By showing the seasonal and annual variations of temperature, salinity, dissolved oxygen and other parameters, readers gain a comprehensive idea about the hydrographic conditions at BE during 2005-2017. We rewrite the lines 86-87 which read now: There is no significant river runoff to Eckernförde Bay. Hence, the hydrographical conditions

are mainly dominated by saline water input from the North Sea and less saline water from the Baltic Proper, which is typical for that region.

Specific comments: -Lines 129-130: How did you shifted the data to the 15th? Include procedure and assumptions in the text.

Reply: Since the time of sampling varied every month (usually 20-40 days interval), it would be easier for comparison and data analysis if all the samples were collected with a regular spacing. In this case, we ignored the slight time difference and assumed that all of the samples were collected on the same day every month. We have replaced lines129-130 with the following sentence: Sampling time varied for every month (usually 20-40 day interval), but for the statistical analysis, data was assumed to be regularly spaced as differences on weekly scales were minor.

Line 170: Could you explicitly explain in the text how did you computed Sc, instead just give the reference? What is the equation for computing Sc?

Reply: Sc was computed as: $Sc = v/D\_N2O$

$D\_N2O = 3.16 \times 10^{(-6)} e^{(-18370/RT)}$

where v is the kinematic viscosity of seawater, which is calculated from the empirical equations given in Siedler and Peters (1986), and DN2O is the diffusion coefficient of N2O in seawater. R is the universal gas constant and T is the water temperature in K. We used the DN2O from Rhee (2000). We have incorporated this part into section 2.4.

Lines 176-184: The comparison of the range of concentrations found between Boknis and other time-series would be better move later in the text, since the reader at this point does not have enough information about the causes that differentiate it from other time-series. The authors should better discuss not only the different magnitudes of the N2O concentrations, but also the site-specific processes responsible of such differences.

Reply: Thank you for the suggestion. The purpose of the comparison is just to give a

general idea about the values of the few time-series N2O measurement published so far, because it might be different from the normal cruises which only last for days or months. A comparison of N2O concentrations between different time-series analysis is not the major topic of the manuscript, and a discussion about the site-specific processes requires detailed information on the environmental variability of the time-series stations, which does not fit in the scope of the manuscript. In this case, we keep this part in section 3.1.

Lines 207...: In case there is additional information in the BTS, such as chlorophyll, during the study period, show the data in figures instead to refer to previous studies.

Reply: Chlorophyll a data were added and are now shown in a new supplement.

Lines 235: are there NH4 data available at the study site during the study period? In that case, it would be better to show them for the discussion instead to appeal to a reference.

Reply: NH4+ data were added and are now shown in a new supplement.

Lines 238-239: "Denitrification is inhibited by the presence of O2 and thus nitrification is presumably responsible for the high N2O concentrations in winter/early spring." This statement is not correct at all. The production of N2O by denitrification can occur at suboxic and hypoxic environments. Please, modify this sentence.

Reply: Thank you for pointing out the problem. We modified the text to "Denitrification is inhibited by the presence of high concentrations of dissolved O2 (> 20 $\mu$mol L-1) and..."

Line 239-240: The authors should normalized the N2O and pH to a constant temperature. Otherwise, temperature changes can be the responsible of this relationship because of thermodynamics changes and not necessarily due to nitrification. In fact, it is not as clear the positive correlation between the N2O and pH in figure 5, since for pH higher than 7.6, there is no apparent trend between N2O and pH. The relationship between pH and N2O obtained during incubations experiments described by Rees et al. (2016) cannot be directly compare to this study, since the experimental conditions and approaches are completely different. The authors should rewrite the entire paragraph.

Reply: We need to delete this part from the manuscript because after double-checking the data, we realize that some of the pH values were not calibrated properly. After re-calibration, the relationship between N2O and pH no long exists. We are very sorry about the mistake.

Lines 263-269: Again, the temperature salinity and Chla information at Boknis are mentioned in the text, but data are missed. If data for these parameters exist, the authors should include in the manuscript. It would reinforce some of the statement that now could look only speculative.

Reply: Temperature, salinity and Chlorophyll a data were added and are now shown in a new supplement.

Lines 287-288: "Although the observed temperatures and salinities during October 2016–April 2017 were comparable to other years,..". Please, show temperature and salinity.

Reply: Data is now shown in the supplement (see replies above).

Lines 295-296: "Considering the classical view of N2O consumption via denitrification under hypoxic and anoxic conditions". This is contrary to the statement done at lines 238-239. Consider to rewrite the first one.

Reply: Thank you pointing this out. We have revised the first one as suggested.

Lines 304-306; 308-309: The authors should make use of temperature, salinity or density to show changes in water masses.

Reply: Temperature and salinity data can be found in the supplement. Mixed layer variations can be seen in Fig. 4. As is mentioned above, it is difficult to show the changes

in water masses. We suggested that the low-N2O water was a result of advection because vertical exchange can be excluded. However, we do not have any evidence since we did not measure dissolved N2O from adjacent waters.

Lines 313: Instead of "presence" it would be more correct "concentration/level"

Reply: We have revised it as suggested.

Lines 320: "We did not observe an exceptional spring algae bloom in 2017". Please, consider to include Chla or POM to support this statement.

Reply: Chlorophyll a data is now shown in the supplement (see replies above). Unfortunately, POM data are not available. Secchi depth, a proxy of water transparency, is slightly lower in March 2017 than the average value. This could be used to support the statement. We modified the text to "Secchi depth, a proxy of water transparency, was 3.8 m in March 2017, which is only slightly lower compared to the monthly average value for March (4.5±1.8 m). There was no exceptional spring algae bloom and thus we infer that assimilative uptake of nutrients by phytoplankton was not responsible for the low nutrients concentrations."

Lines 319: Why can not be shown the Chla data?

Reply: Chlorophyll a data is now shown in the supplement.

Lines 331-335: The author should also discuss the potential dependence of rates on temperature and its impact on the seasonal variations of N2O production/consumption trough the text.

Reply: Unfortunately, we did not measure N2O production/consumption rates at BE. A discussion about the potential temperature dependence of rates is, thus, too speculative. Besides, there is no significant temperature anomaly during the low-N2O-event. In this case, we suggest that the impact of temperature on the low-N2O event could be excluded.

Lines 356-357: Please, consider to support this statement with the salinity data 371-373. Please, show the density (or temperature and salinity) record to track the upwelling event in autumn 2017. Lines 377-378: Please, show the chlorophyll data. Lines385_386: Please, show the ammonium data.

Reply: The data are now shown in the supplement.

Lines 394-399: This is a very speculative paragraph as it is written. Could you give any evidence for these potential explanations of the homogeneous distribution of N2O?

Reply: Although the oxic/anoxic interface, where enhanced N2O production occurs, lasts for several months, the high N2O concentrations were usually observed only in late autumn (Fig. 4). We agree this paragraph is speculative. There are just some "potential explanations" for why there is no enhanced N2O in early autumn. Unfortunately, we do not have any further evidence to support the conjecture.

-Section 3.5: The author should evaluate in the results the impact of the dissolved gas analysis uncertainty in the air-sea flux computation and the uncertainty introduced in the net seasonal and annual air-sea NO fluxes.

Reply: The uncertainty of flux density, which is mainly derived from KN2O, with a minor contribution from the error of trace gas analysis, was estimated to be 20% (Wanninkhof, 2014). The average flux density at BE was $3.5 \pm 12.4$ $\mu$mol m-2 d-1. With a large uncertainty in the flux density, it is difficult to compute meaningful seasonal/annual fluxes. In this case, we only discussed the variation of flux density in section 3.5. We have added the uncertainty of flux density in section 3.5.

Lines 416-424: The authors show that N2O concentration change seasonally, but the saturation stay almost constant. So, how can the author affirm that emissions are controlled by temperature?

Reply: There is a seasonality in surface N2O concentrations but not for the N2O saturation. During the transformation of concentrations into saturations, the effect of temperature on the saturation is more essential than the effect of salinity. In summer when surface N2O concentrations are low, N2O saturations are increased by the relative high temperature (because the equilibrium concentration is decreased). In winter the N2O concentrations are high, but N2O saturations are decreased because of the high N2O solubility at low temperature condition. Temperature is "buffering" the variation of saturation and thus affecting N2O emissions, and this is our point of "a modulating role". To clarify this point we rewrite the paragraph: We found a weak seasonal cycle for surface N2O concentrations, with high N2O concentrations occurring in winter/early spring and low concentrations occurring in summer/autumn, but no such cycle for N2O saturation. The seasonality in concentration but not in saturation could be largely attributed to the effect of temperature on N2O solubility: In summer when surface N2O concentrations are low, N2O saturations are increased by the relative high temperature; and vice versa in winter. Although salinity also affects N2O solubility, its contribution is negligible compared to temperature. Temperature alleviated the fluctuation of surface N2O saturation and thus affected the sea-to-air N2O fluxes. We conclude that temperature plays a modulating role for N2O emissions.

Lines 476-484: Unless the author do not include salinity and temperature, they should not used them to conclude the hydrographic conditions at Boknis Eck. Further studies about the hydrography at the BE would complete the picture together with the biogeochemical data at the BE time-series station.

Reply: We agree that changing hydrographic conditions will affect N2O cycling at Boknis Eck as well. This, however, will require modeling studies which need to take into account the ongoing environmental changes of temperature, deoxygenation, changing frequency of North Sea water inflow etc. We think that a detailed discussion of potential future projections of the environmental variability of Boknis Eck/Eckernförde Bay is beyond the scope of the manuscript.

Please also note the supplement to this comment:

https://www.biogeosciences-discuss.net/bg-2019-165/bg-2019-165-AC1-supplement.pdf

---

## Author Comment (AC2) · 19 Aug 2019

Thank you very much for the comments. They are thought-provoking and helpful to improve our manuscript.

Quantifying the concentrations and dynamics of dissolved N2O in seawater is important for understanding the climate change, but conducting measurements of sufficient duration to determine trends over seasonal, interannual, and decadal time frames for any marine ecosystem remains a challenging task. A long-term Time-Series Station like Boknis Eck in the Eckernförde Bay can provide invaluable information for documenting the role of oceans in relation to N2O, hence this type of study is significant

for our scientific understanding. The paper is well-written and clear. I only have few comments/suggestions below:

Lines 95-96: more information is needed on seawater sample collection.

Reply: The sampling procedure is described in detail in lines 95-100 on page 3. Thus we do not see a need to revise the text.

Lines 108-109: The N2O concentration of standard gases should be provided.

Reply: We have added this information in the method section.

Line 154-155: : : :dry mole fractions of atmospheric N2O at the time of the sampling. This description is not the fact since atmospheric N2O at the time of the sampling has not been measured and the monthly average of N2O data measured at Mace Head was used to N2Oeq.

Reply: Thank you for pointing this out. We modified the sentence. It reads now: "Since the atmospheric N2O mole fractions were not measured at the BE Time-Series Station, atmospheric dry mole fractions of N2O were derived from the monthly average of N2O data at Mace Head, Ireland (AGAGE, http://agage.mit.edu/), instead."

Lines 185-193: NH4+ concentrations should be provided in the text as well as Figure 2.

Reply: Both reviewers requested to show additional temperature, salinity, chlorophyll a and ammonium data. These data were not explicitly shown in our ms because they have been published already in Lennartz et al.: Long-term trends at the Boknis Eck time series station (Baltic Sea), 1957–2013: does climate change counteract the decline in eutrophication? Biogeosciences, 11, 6323–6339, https://doi.org/10.5194/bg-11-6323-2014.). However, in order to provide this obvious lacking information, we decided to include the seasonal and annual variations of temperature, salinity, chlorophyll a and ammonium in a new supplement to our manuscript.

Lines 221-222: The expression caused misunderstanding.

Reply: We would like to change it to "The seasonal variations of NO2- and NO3- were significantly correlated with each other ([NO3−]=11.59[NO2−]-0.51, R2=0.80, n=72, p<0.0001) and high concentrations were observed for both in winter."

Lines 239-243: Does N2O correlate with NH4+?

Reply: There is no straightforward relationship between N2O and NH4+. By the way, we realized that there were some problems with the calibration of pH data and, therefore, this part will be deleted.

Line 462: The year of 2015 should be 2005?

Reply: Thank you for pointing out the mistake. It should be 2005, which is the beginning of the N2O measurement.

Line 476: There is no temperature data provided at all in this study but with a conclusion 'Temperature plays a modulating role for the N2O emission at the BE Time-Series Station'. I suggest to provide t data in Figure 2 and more t data provided in related discussion in the text

Reply: We would like to show temperature data in the supplement. Also will rewrite the relevant text in section 3.5 to explain how temperature modulates N2O emissions in. See our reply to reviewer #1 as well.

Figure 2: It would be better for the authors to provide the vertical profiles of t, s, density, NH4+ and Chl a.

Reply: The data is now shown in the supplement.

Figure 6: The vertical profiles of hydrological parameters, such as t, s and density are needed to help understand the possible influence of physical processes on N2O distribution as discussed between lines 301 and 309.

Reply: Vertical profiles of temperature, salinity, NH4+ and Chl a are shown in supplement. Mixed layer variations can be seen in Fig. 4.

Figure 8: title is needed for x- and y-axis at figure b and d Figure 9: title is needed for x- and y-axis

Reply: We have added new titles in the figures.

Please also note the supplement to this comment:
https://www.biogeosciences-discuss.net/bg-2019-165/bg-2019-165-AC2-supplement.pdf

―――――――――――――――――――

**Supplement:**

*Supplement of*

**A multi-year observation of nitrous oxide at the Boknis Eck Time-Series Station in the Eckernförde Bay (southwestern Baltic Sea)**

Xiao Ma, Sinikka T. Lennartz, and Hermann W. Bange

*Correspondence to:* Xiao Ma (mxiao@geomar.de)

[Figure]

Supplementary Figure 1. Vertical distributions of temperature, salinity, chlorophyll *a* and $NH_4^+$ from the BE Time-Series Station during 2005–2017. Please note that high $NH_4^+$ concentrations (>20 μM/L, which are detected in the bottom water of October 2005, May 2006, October 2013, October 2014, October and November 2016) were removed for better visualization.

[Figure]

Supplementary Figure 2. Average vertical distributions of temperature, salinity, chlorophyll *a* and NH$_4^+$ from the BE Time-Series Station during 2005–2017

---

## Author Response (AR1)

**Reply to reviewer #1**

*General comments:*
*The present paper examined the seasonal and annual variations of dissolved N2O in a time-series station located in the southwestern Baltic Sea. The results show the coupled variations between the N2O anomalies, the oxygen concentrations, and nutrients.*
*The paper presents a valuable new dataset of N2O and related biogeochemical parameters in a marine region subject to extensive human activities and so nutrients inputs, responsible of the deoxygenation in the Baltic Sea. After the revision, I consider that the manuscript is highly interesting and provide relevant information about processes occurring to the N2O in the Boknis Eck. The paper is well written and structured, with an appropriate description of the state of the art, objectives are clearly outlined and discussion precisely referenced. The main strength of the paper is the monthly sampling undertaken during twelve years.*
*However, there are several weaknesses in the paper. First, the authors make the discussion of the results based on data that are included in this study. There are references to data that exist but are not shown. But, in other to discuss about upwelling and hydrographic changes, about algal blooms and ammonium changes, the salinity, temperature, chlorophyll and ammonium data should be included and shown in this paper.*
Both reviewers requested to show additional temperature, salinity, chlorophyll *a* and ammonium data. These data were not explicitly shown in our manuscript because they have been published already in Lennartz et al.: Long-term trends at the Boknis Eck time series station (Baltic Sea), 1957–2013: does climate change counteract the decline in eutrophication? Biogeosciences, 11, 6323–6339, https://doi.org/10.5194/bg-11-6323-2014. However, in order to provide this obvious lacking information, we decided to include the seasonal and annual variations of temperature, salinity, chlorophyll *a* and ammonium in a new supplement to our manuscript.

*Secondly, the paper lack of a proper description of the water masses presents at BE and their temporal variability.*
Considering that all the measurements were conducted at one fixed location, it is difficult to investigate the water masses based on our data. However, the hydrological condition at BE are not complicated. As is written in Lines 86-87, there is no pronounced river input, and saline water from the North Sea plays a dominant role. By showing the seasonal and annual variations of temperature, salinity, dissolved oxygen and other parameters, readers gain a comprehensive idea about the hydrographic conditions at BE during 2005-2017.
We rewrite the lines 86-87 which read now:
There is no significant river runoff to Eckernförde Bay. Hence, the hydrographical conditions are mainly dominated by saline water input from the North Sea and less saline water from the Baltic Proper, which is typical for that region.

*Specific comments:*
*-Lines 129-130: How did you shifted the data to the 15th? Include procedure and assumptions in the text.*

Since the time of sampling varied every month (usually 20-40 days interval), it would be easier for comparison and data analysis if all the samples were collected with a regular spacing. In this case, we ignored the slight time difference and assumed that all of the samples were collected on the same day every month.

We have replaced lines129-130 with the following sentence: Sampling time varied for every month (usually 20-40 day interval), but for the statistical analysis, data was assumed to be regularly spaced as differences on weekly scales were minor.

*Line 170: Could you explicitly explain in the text how did you computed Sc, instead just give the reference? What is the equation for computing Sc?*

Sc was computed as:

$$Sc = v/D_{N2O}$$

$$D_{N2O} = 3.16 \times 10^{-6} e^{-18370/RT}$$

where $v$ is the kinematic viscosity of seawater, which is calculated from the empirical equations given in Siedler and Peters (1986), and $D_{N2O}$ is the diffusion coefficient of $N_2O$ in seawater. $R$ is the universal gas constant and $T$ is the water temperature in K. We used the $D_{N2O}$ from Rhee (2000).

We have incorporated this part into section 2.4.

*Lines 176-184: The comparison of the range of concentrations found between Boknis and other time-series would be better move later in the text, since the reader at this point does not have enough information about the causes that differentiate it from other time-series. The authors should better discuss not only the different magnitudes of the N2O concentrations, but also the site-specific processes responsible of such differences.*

Thank you for the suggestion. The purpose of the comparison is just to give a general idea about the values of the few time-series $N_2O$ measurement published so far, because it might be different from the normal cruises which only last for days or months. A comparison of $N_2O$ concentrations between different time-series analysis is not the major topic of the manuscript, and a discussion about the site-specific processes requires detailed information on the environmental variability of the time-series stations, which does not fit in the scope of the manuscript. In this case, we keep this part in section 3.1.

*Lines 207…: In case there is additional information in the BTS, such as chlorophyll, during the study period, show the data in figures instead to refer to previous studies.*

Chlorophyll *a* data were added and are now shown in a new supplement.

*Lines 235: are there NH4 data available at the study site during the study period? In that case, it would be better to show them for the discussion instead to appeal to a reference*

$NH_4^+$ data were added and are now shown in a new supplement.

*Lines 238-239: "Denitrification is inhibited by the presence of O2 and thus nitrification is presumably responsible for the high N2O concentrations in winter/early spring." This statement is not correct at all. The production of N2O by denitrification can occur at suboxic and hypoxic environments. Please, modify this sentence.*

Thank you for pointing out the problem. We modified the text to "Denitrification is inhibited by the presence of high concentrations of dissolved $O_2$ (> 20 µmol $L^{-1}$) and…"

*Line 239-240: The authors should normalized the N2O and pH to a constant temperature. Otherwise, temperature changes can be the responsible of this relationship because of thermodynamics changes and not necessarily due to nitrification. In fact, it is not as clear the positive correlation between the N2O and pH in figure 5, since for pH higher than 7.6, there is no apparent trend between N2O and pH. The relationship between pH and N2O obtained during incubations experiments described by Rees et al. (2016) cannot be directly compare to this study, since the experimental conditions and approaches are completely different. The authors should rewrite the entire paragraph.*
We need to delete this part from the manuscript because after double-checking the data, we realize that some of the pH values were not calibrated properly. After re-calibration, the relationship between $N_2O$ and pH no long exists. We are very sorry about the mistake.

*Lines 263-269: Again, the temperature salinity and Chla information at Boknis are mentioned in the text, but data are missed. If data for these parameters exist, the authors should include in the manuscript. It would reinforce some of the statement that now could look only speculative.*
Temperature, salinity and Chlorophyll *a* data were added and are now shown in a new supplement.

*Lines 287-288: "Although the observed temperatures and salinities during October 2016–April 2017 were comparable to other years,..". Please, show temperature and salinity.*
Data is now shown in the supplement (see replies above).

*Lines 295-296: "Considering the classical view of N2O consumption via denitrification under hypoxic and anoxic conditions". This is contrary to the statement done at lines 238-239. Consider to rewrite the first one.*
Thank you pointing this out. We have revised the first one as suggested.

*Lines 304-306; 308-309: The authors should make use of temperature, salinity or density to show changes in water masses.*
Temperature and salinity data can be found in the supplement. Mixed layer variations can be seen in Fig. 4. As is mentioned above, it is difficult to show the changes in water masses. We suggested that the low-$N_2O$ water was a result of advection because vertical exchange can be excluded. However, we do not have any evidence since we did not measure dissolved $N_2O$ from adjacent waters.

*Lines 313: Instead of "presence" it would be more correct "concentration/level"*
We have revised it as suggested.

*Lines 320: "We did not observe an exceptional spring algae bloom in 2017". Please, consider to include Chla or POM to support this statement.*

Chlorophyll a data is now shown in the supplement (see replies above). Unfortunately, POM data are not available. Secchi depth, a proxy of water transparency, is slightly lower in March 2017 than the average value. This could be used to support the statement.

We modified the text to "Secchi depth, a proxy of water transparency, was 3.8 m in March 2017, which is only slightly lower compared to the monthly average value for March (4.5±1.8 m). There was no exceptional spring algae bloom and thus we infer that assimilative uptake of nutrients by phytoplankton was not responsible for the low nutrients concentrations."

*Lines 319: Why can not be shown the Chla data?*
Chlorophyll *a* data is now shown in the supplement.

*Lines 331-335: The author should also discuss the potential dependence of rates on temperature and its impact on the seasonal variations of N2O production/consumption trough the text.*
Unfortunately, we did not measure $N_2O$ production/consumption rates at BE. A discussion about the potential temperature dependence of rates is, thus, too speculative. Besides, there is no significant temperature anomaly during the low-$N_2O$-event. In this case, we suggest that the impact of temperature on the low-$N_2O$ event could be excluded.

*Lines 356-357: Please, consider to support this statement with the salinity data 371-373. Please, show the density (or temperature and salinity) record to track the upwelling event in autumn 2017. Lines 377-378: Please, show the chlorophyll data. Lines385_386: Please, show the ammonium data.*
The data are now shown in the supplement.

*Lines 394-399: This is a very speculative paragraph as it is written. Could you give any evidence for these potential explanations of the homogeneous distribution of N2O?*
Although the oxic/anoxic interface, where enhanced $N_2O$ production occurs, lasts for several months, the high $N_2O$ concentrations were usually observed only in late autumn (Fig. 4). We agree this paragraph is speculative. There are just some "potential explanations" for why there is no enhanced $N_2O$ in early autumn. Unfortunately, we do not have any further evidence to support the conjecture.

*-Section 3.5: The author should evaluate in the results the impact of the dissolved gas analysis uncertainty in the air-sea flux computation and the uncertainty introduced in the net seasonal and annual air-sea NO fluxes.*
The uncertainty of flux density, which is mainly derived from $K_{N2O}$, with a minor contribution from the error of trace gas analysis, was estimated to be 20% (Wanninkhof, 2014). The average flux density at BE was 3.5±12.4 $\mu mol\ m^{-2}\ d^{-1}$. With a large uncertainty in the flux density, it is difficult to compute meaningful seasonal/annual fluxes. In this case, we only discussed the variation of flux density in section 3.5.

We have added the uncertainty of flux density in section 3.5.

*Lines 416-424: The authors show that N2O concentration change seasonally, but the saturation stay almost constant. So, how can the author affirm that emissions are controlled by temperature?*

There is a seasonality in surface $N_2O$ concentrations but not for the $N_2O$ saturation. During the transformation of concentrations into saturations, the effect of temperature on the saturation is more essential than the effect of salinity. In summer when surface $N_2O$ concentrations are low, $N_2O$ saturations are increased by the relative high temperature (because the equilibrium concentration is decreased). In winter the $N_2O$ concentrations are high, but $N_2O$ saturations are decreased because of the high $N_2O$ solubility at low temperature condition. Temperature is "buffering" the variation of saturation and thus affecting $N_2O$ emissions, and this is our point of "a modulating role".

To clarify this point we rewrite the paragraph:

We found a weak seasonal cycle for surface $N_2O$ concentrations, with high $N_2O$ concentrations occurring in winter/early spring and low concentrations occurring in summer/autumn, but no such cycle for $N_2O$ saturation. The seasonality in concentration but not in saturation could be largely attributed to the effect of temperature on $N_2O$ solubility: In summer when surface $N_2O$ concentrations are low, $N_2O$ saturations are increased by the relative high temperature; and vice versa in winter. Although salinity also affects $N_2O$ solubility, its contribution is negligible compared to temperature. Temperature alleviated the fluctuation of surface $N_2O$ saturation and thus affected the sea-to-air $N_2O$ fluxes. We conclude that temperature plays a modulating role for $N_2O$ emissions.

*Lines 476-484: Unless the author do not include salinity and temperature, they should not used them to conclude the hydrographic conditions at Boknis Eck. Further studies about the hydrography at the BE would complete the picture together with the biogeochemical data at the BE time-series station.*

We agree that changing hydrographic conditions will affect $N_2O$ cycling at Boknis Eck as well. This, however, will require modeling studies which need to take into account the ongoing environmental changes of temperature, deoxygenation, changing frequency of North Sea water inflow etc. We think that a detailed discussion of potential future projections of the environmental variability of Boknis Eck/Eckernförde Bay is beyond the scope of the manuscript.

**Reply to reviewer #2**

*Quantifying the concentrations and dynamics of dissolved N2O in seawater is important for understanding the climate change, but conducting measurements of sufficient duration to determine trends over seasonal, interannual, and decadal time frames for any marine ecosystem remains a challenging task. A long-term Time-Series Station like Boknis Eck in the Eckernförde Bay can provide invaluable information for documenting the role of oceans in relation to N2O, hence this type of study is significant for our scientific understanding. The paper is well-written and clear. I only have few comments/suggestions below:*

*Lines 95-96: more information is needed on seawater sample collection.*
The sampling procedure is described in detail in lines 95-100 on page 3. Thus we do not see a need to revise the text.

*Lines 108-109: The N2O concentration of standard gases should be provided.*
We have added this information in the method section.

*Line 154-155: : : :dry mole fractions of atmospheric N2O at the time of the sampling. This description is not the fact since atmospheric N2O at the time of the sampling has not been measured and the monthly average of N2O data measured at Mace Head was used to N2Oeq.*
Thank you for pointing this out. We modified the sentence. It reads now: "Since the atmospheric $N_2O$ mole fractions were not measured at the BE Time-Series Station, atmospheric dry mole fractions of $N_2O$ were derived from the monthly average of $N_2O$ data at Mace Head, Ireland (AGAGE, http://agage.mit.edu/), instead."

*Lines 185-193: NH4+ concentrations should be provided in the text as well as Figure 2.*
Both reviewers requested to show additional temperature, salinity, chlorophyll *a* and ammonium data. These data were not explicitly shown in our manuscript because they have been published already in Lennartz et al.: Long-term trends at the Boknis Eck time series station (Baltic Sea), 1957–2013: does climate change counteract the decline in eutrophication? Biogeosciences, 11, 6323–6339, https://doi.org/10.5194/bg-11-6323-2014. However, in order to provide this obvious lacking information, we decided to include the seasonal and annual variations of temperature, salinity, chlorophyll *a* and ammonium in a new supplement to our manuscript.

*Lines 221-222: The expression caused misunderstanding.*
We would like to change it to "The seasonal variations of $NO_2^-$ and $NO_3^-$ were significantly correlated with each other ($[NO_3^-]=11.59[NO_2^-]-0.51$, $R^2=0.80$, n=72, p<0.0001) and high concentrations were observed for both in winter.
"

*Lines 239-243: Does N2O correlate with NH4+?*
There is no straightforward relationship between $N_2O$ and $NH_4^+$. By the way, we realized that there were some problems with the calibration of pH data and, therefore, this part will be deleted.

*Line 462: The year of 2015 should be 2005?*
Thank you for pointing out the mistake. It should be 2005, which is the beginning of the $N_2O$ measurement.

*Line 476: There is no temperature data provided at all in this study but with a conclusion 'Temperature plays a modulating role for the N2O emission at the BE Time-Series Station'. I suggest to provide t data in Figure 2 and more t data provided in related discussion in the text*
We would like to show temperature data in the supplement. Also will rewrite the relevant text in section 3.5 to explain how temperature modulates $N_2O$ emissions in. See our reply to reviewer #1 as well.

*Figure 2: It would be better for the authors to provide the vertical profiles of t, s, density, NH4+ and Chl a.*
The data is now shown in the supplement.

*Figure 6: The vertical profiles of hydrological parameters, such as t, s and density are needed to help understand the possible influence of physical processes on N2O distribution as discussed between lines 301 and 309.*
Vertical profiles of temperature, salinity, $NH_4^+$ and Chl *a* are shown in supplement. Mixed layer variations can be seen in Fig. 4.

*Figure 8: title is needed for x- and y-axis at figure b and d*
*Figure 9: title is needed for x- and y-axis*

We have added new titles in the figures.

According to the comments from the reviewers, the following changes are made in the manuscript:

1. Show seasonal and annual variations of temperature, salinity, chlorophyll *a* and ammonium in a new supplement to our manuscript.
2. Rewrite the lines 86-88.
3. Add more details about standard gases in line 109.
4. Rewrite the lines 129-131.
5. Rewrite the lines 156-158.
6. Add more details in *Sc* computation in Section 2.4 (lines 165-171).
7. Rewrite the lines 221-223.
8. Rewrite the lines 238-239.
9. Delete the discussion about the relationship between $N_2O$ and pH in section 3.2, and the corresponding figure was removed as well.
10. Modify the sentence in line 309.
11. Rewrite the lines 316-320.
12. Rewrite the lines 414-422.
13. Add the uncertainty of flux density computation in lines 428-429.
14. Add new titles in Fig. 7 and 8.

Besides the changes mentioned above, we also revised:

1. The computation of $\Delta N_2O$ and AOU were removed from section 2.4 because the corresponding discussion was no longer in Results and Discussion.
2. Few typos in the manuscript were corrected.

[revised manuscript text omitted]

---

## Author Response (AR2)

Reply to report #1

Thank you for your comments.

The authors have incorporated most of the comments of the reviewers, and paper now is greatly improved with the incorporation of salinity, temperature, Chl a and NH4. I commend the authors on their efforts, and find that they have responded appropriately to the reviews. I had just a few minor comments for consideration.

Lines 130-131: I agree with the authors that sample date should be adjusted to a regular spacing. However, I do not agree with the assumption of no not temporal variability between weeks. The authors should use a lineal interpolation (or test other models) to homogenize dates, or in lack of this, demonstrate that the uncertainty introduced with such assumption is not significant.

We used a linear interpolation to homogenize the dates and found a overall error of 4.2%. In this case, we modified the text into "Sampling time varied for every month (usually 20-40 day interval), but for the statistical analysis, data was assumed to be regularly spaced as the uncertainty introduced was not significant (<5%)."

Lines 238-239: The authors should include a reference for the O2 limit used for the inhibition of denitrification.
According to Tiedje (1988), we included the $O_2$ threshold of ~10 µmol $L^{-1}$ for the inhibition of denitrification.

The following changes have been made in the manuscript:

1. Include the uncertainty introduced from the date shift in line 131.
2. Include the $O_2$ threshold for the inhibition of denitrification in lines 240–241.
3. Add a reference in lines 651–653.

[revised manuscript text omitted]